# Non-telecentric two-photon microscopy for 3D random access mesoscale imaging

F. K. Janiak [1✉], P. Bartel[1], M. R. Bale [1], T. Yoshimatsu [1], E. Komulainen[1], M. Zhou[1], K. Staras [1], L. L. Prieto-Godino [2], T. Euler [3,4], M. Maravall [1] & T. Baden [1,3✉]

Diffraction-limited two-photon microscopy permits minimally invasive optical monitoring of neuronal activity. However, most conventional two-photon microscopes impose significant constraints on the size of the imaging field-of-view and the specific shape of the effective excitation volume, thus limiting the scope of biological questions that can be addressed and the information obtainable. Here, employing a non-telecentric optical design, we present a low-cost, easily implemented and flexible solution to address these limitations, offering a several-fold expanded three-dimensional field of view. Moreover, rapid laser-focus control via an electrically tunable lens allows near-simultaneous imaging of remote regions separated in three dimensions and permits the bending of imaging planes to follow natural curvatures in biological structures. Crucially, our core design is readily implemented (and reversed) within a matter of hours, making it highly suitable as a base platform for further development. We demonstrate the application of our system for imaging neuronal activity in a variety of examples in zebrafish, mice and fruit flies.

[1] Sussex Neuroscience, School of Life Sciences, University of Sussex, Brighton, UK. [2] The Francis Crick Institute, London, UK. [3] Institute of Ophthalmic Research, University of Tübingen, Tübingen, Germany. [4] Centre for Integrative Neuroscience, University of Tübingen, Tübingen, Germany. ✉email: f.k.janiak@sussex.ac.uk; t.baden@sussex.ac.uk

L aser scanning two-photon (2P) microscopy allows the imaging of live cellular processes deep inside intact tissue with high signal-to-noise, temporal fidelity and spatial resolution[1]. Nonetheless, standard diffraction-limited 2P setups with a collimated laser excitation beam have several key characteristics that constrain their broad applicability; namely, a typically small field of view (FOV), a fixed-size excitation spot and restricted options for rapid random access 3-dimensional scans. These are significant limitations because the biological samples that are interrogated with 2P microscopy can exhibit substantial variations in size and spatial structure. For example,

the volume of an adult mouse brain is approximately four orders of magnitude larger than that of a larval zebrafish, and seven orders of magnitude larger than a first instar larval fruit fly (Fig. 1a). Similarly, neuronal sub-structures are also highly variable in density and size, ranging from sub-micron levels for some synapses up to 20 μm or more for some somata. Additionally, neural densities vary by more than an order of magnitude across different animal brains[2]. As such, 2P microscopy tends to reveal very different levels of detail and organization across its diverse experimental applications. To maximize biological information, upgrades for 2P microscopy should enable the imaging of

**Fig. 1 Non-telecentric beam optics in 2-photon microscopy. a** *Drosophila* (left), larval zebrafish (centre) and adult mouse schematics (right) with central nervous system highlighted (green) to illustrate size differences. Insets next to the mouse for size-comparison on the same scale. **b** Optical configurations of standard diffraction limited (DL, left) 2P setup with parallel laser beam entering objective's back aperture. Right, non-telecentric (nTC, middle, right) configurations use a diverging laser beam instead. As a result, the field of view and focal distance are expanded, and the point spread function (PSF) elongates. These effects scale with the angle of divergence (compare $nTC_1$ and $nTC_2$). **c** Typical neuronal somata in species shown in (**a**), as interrogated by 2P setups shown in (**b**), respectively. **d** In vivo 7 *dpf* larval zebrafish (HuC:GCaMP6f) imaged with a Sutter-MOM DL setup at full field of view (top) and when zoomed in to reveal individual neuronal somata (bottom) as indicated. **e** same zebrafish as (**d**), next to two further zebrafish imaged using $nTC_2$ configuration at maximal field of view (top). Zooming in to the same area as in (**d**, bottom) reveals cellular detail (**e**, bottom). **f** In vivo adult mouse cranial window over somatosensory cortex imaged with $nTC_2$ maximal field of view (top) and when zoomed in as indicated (bottom). **g** Left: Optical configuration of a standard DL setup with collimation system consisting of a scan lens and a tube lens to set-up an infinity collimated laser beam at the objective's back aperture. Effective refractive power and relative distances of lenses indicated. The intermediary focal point (IFP) is behind the scan lens (arrowhead). Middle: $nTC_1$ configuration replaces the scan lens with two plano-convex lenses (L1,2). The relative position of L2 to the tube lens defines the position of the IFP, which is now further along the laser path. As a result, the field of view can be expanded to between 1.2 and 1.8 mm. Right: $nTC_2$ configuration using a single plano-convex lens (L3) allows FOV expansion to 2.5–3.5 mm. **h** complete nTC setup, including an ETL positioned in front of the scan mirrors for rapid axial-scanning. PMTs, Photomultipliers. **i** FOV expansion under nTC combines two effects: Increased focal distance (left) and reduced numerical aperture (N.A., right), which together yield a larger effective focal plane and enlarged PSF. **j** Power at sample measured for all configurations, expressed as a percentage of the power that reaches the scanning mirrors. **k** point spread functions (PSFs) measured for all configurations, with size of typical neuronal somata of different species indicated. All scale-bars 10 μm. **l**, **m**, lateral (**l**) and axial (**m**) spread of the PSFs quantified. Errors in mean ± s.d., $n = 20$ experimentally independent samples per measurement. Data leading to **j**, **l**, **m** in Source data file.

neuronal activity from many neural structures of a given size and density across a sufficiently large 3D volume of tissue at sufficiently high frame rates for the chosen neuronal process and biosensor.

In response to this demand, a profusion of custom modifications to 2P microscopes have been developed to expand the spatial and temporal boundaries over which neural structures can be optically interrogated. For example, the maximal planar field of view (FOV) has been increased from typically 0.7 mm to between 3.1 and 10 mm diameter by the exchange (3.1 mm: ref. [3], 7 mm: ref. [4]) or size-increase of optical components (10 mm)[5], custom built objectives (3.1 mm)[6], enhanced scan engines (5 mm)[7] and a mesoscope configuration (5 mm)[8] to allow 'mesoscale' interrogation of neural circuits. In parallel, customizations using multiple beams have allowed simultaneous scanning of distant brain regions[6,9,10]. Likewise, higher temporal resolutions have been achieved by tailoring the point spread function (PSF) to the geometry and distribution of the neuronal structures of interest, thus increasing signal-to-noise ratio (SNR) and, in turn, decreasing the minimally-required dwell time per pixel[11,12]. Moreover, the imaging plane has been axially expanded by engineering an excitation spot with Bessel focus[13,14] or by elongated Gaussian foci stereoscopy[15]. These customizations provide efficient ways to merge image structures that are located at different depths into a single volumetric plane. Furthermore, in recent years, systems integrating acousto-optic deflectors[16,17], electrical tunable lenses[9,18–21] and remote focusing units[8,14] have enabled quasi-simultaneous multiplane volumetric scans.

These types of extensions have been essential in driving the field forward, yet many are expensive, require custom-produced optical elements, complex optical alignment and/or introduce new limitations. The latter can include limitations in both excitation (e.g. power loss[8], wavefront dispersion[17]) and collection[4,5]. Here we introduce an alternative design for 2P microscopy that overcomes many of these limitations while simultaneously approaching the capabilities of a wide range of state-of-the-art performance customisations, and being ultra-low-cost, simple and flexible.

Our non-telecentric (nTC) design, implemented for ~£1000 on an existing Sutter MOM-type 2P setup equipped with a standard ×20 objective, allows the expansion of the planar FOV from typically ~0.7 mm in diameter to anywhere up to 3.5 mm to flexibly suit experimental needs (Fig. 1). This expansion is accompanied by a moderate increase in the system's 3D PSF.

For example, unlike a standard diffraction limited (DL) setup (left in Fig. 1b, c), our nTC setup (right in Fig. 1b, c) allows simultaneous imaging of three entire zebrafish brains (Fig. 1e), or about a third of the width of a mouse's brain (Fig. 1f, Supplementary Movie S1). The addition of an electrically tunable lens (ETL) then allows near-simultaneous sampling in distant brain regions separated in 3 dimensions. Crucially, our solution is both comparatively low-cost and easy to implement without the need for complex optical calibration, thus facilitating its widespread adoption in the community. We anticipate that others will be able to build on our core optical design using existing and new modifications to further increase its capability in the future. We demonstrate the current performance of our system with a range of examples from zebrafish, mice and fruit flies.

## Results

**Non-telecentric optics for field of view expansion.** In traditional laser scanning 2P microscopy (left in Fig. 1b, c, g), a diffraction limited (DL) PSF is generated to excite fluorophores in a typically sub-micron volume of tissue. Here, xy-scanning mirrors reflect the laser beam into a collimation system comprised of a scan and a tube lens. The collimated beam then enters the back aperture of a high numerical aperture (N.A.) objective[22,23] to converge at parallel rays into a DL spot at focal distance[24]. The Gaussian shape of the excitation beam dictates that it is not possible to perfectly match beam width to the objective's back aperture. Instead, the back aperture is typically overfilled with a factor of 1/$e^2$ as a compromise between maximising spatial resolution (i.e. small PSF size) and power transmission[25].

In contrast, our nTC design (middle and right in Fig. 1b, c, g) illuminates the objective's back-aperture with a decollimated and divergent beam. This leads to an increased angle of view as the light exits the objective's front aperture, such that the same angular scan-mirror movement leads to a larger absolute shift in the image plane—thereby greatly increasing the FOV. In parallel, this also alters the effective excitation numerical aperture (N.A.) to yield a larger-than-DL excitation spot (i.e. an elongated PSF) at greater focal distance. The magnitudes of each of these effects scale with the angle of divergence as the beam enters the back aperture of the objective. Accordingly, simply shifting the plano-convex lenses up or down the laser path, or switching between different refractive power lenses, provides for easy control over the system's optical properties.

In the following we show that the use of nTC in 2P microscopy brings about important advantages over the traditional, collimated and diffraction limited (DL) design:

1. The total field of view (FOV) can be expanded several-fold to suit the user's needs.
2. Scan-mirror movements translate into correspondingly larger xy-shifts in the image plane, meaning that even multi-millimetre random access jumps can be achieved with millisecond precision.
3. The addition of an electrically tunable lens (ETL) in front of the scan mirrors allows for similarly extensive expansion in the axial dimension.
4. The simplified optical path (i.e. fewer individual lenses between the scan mirrors and the sample) and under-filling of the objective's back aperture means more laser power is available at the sample plane.
5. The increased working distance provides additional space for access to the preparation, for example with electrodes or stimulation equipment.

We first discuss the required optical modifications and their impact on key excitation parameters (Figs. 1–4, Figs. S1–3), before presenting a series of key use cases of different configurations for the interrogation of neural structure and function across diverse models (Figs. 5–10, Figs. S4, 5).

**A simple scan-lens modification yields up to 7-fold FOV expansion.** An off-the-shelf infinity-corrected galvo-galvo Sutter-MOM setup equipped with a ×20 objective (Zeiss Objective W "Plan-Apochromat" ×20/1.0) offers a square FOV diameter of ~0.5 mm (left in Fig. 1g). This can be principally extended to a round ~0.7 mm diameter usable diffraction limited (DL) FOV of the objective (see below), for example by redesigning the native beam path to allow a greater range of laser travel. Alternatively, however, when underfilling the back aperture of the objective with a diverging laser (middle and right in Fig. 1g), the beam exits the objective front aperture at increasingly obtuse angles at an effectively decreased N.A. (Fig. 1i, Fig. S1a) and comes into focus at a greater distance (Fig. 1g, Fig. S1b). Together, this expands the effective excitation FOV in both xy (increased angle and decreased N.A.) and z (elongated PSF). To achieve this effect, it is necessary to bring the collimated laser beam, having passed the scan mirrors, to an "early" intermediary focal point (IFP) prior to reaching the objective, thus setting up the diverging beam thereafter (Fig. 1g, arrowheads). The specific divergence angle as the beam enters the back-aperture of the objective, which depends on IFP, defines the magnitude of the above-mentioned effects. We present two simple optical solutions (nTC$_1$ and nTC$_2$) to set-up an early IFP and thus expand the effective FOV to varying degrees.

In the standard DL configuration, the scan-lens (SL) and tube lens (TL) are separated from each other at a distance that is equal to their combined focal lengths ($50_{SL} + 200_{TL}$ mm = 250 mm) to collimate the beam (left, Fig. 1g). In nTC$_1$, we removed SL and instead inserted two off-the-shelf plano-convex lenses (L1, modified VISIR 1534SPR136, Leica; L2, LA1229 Thorlabs) with focal lengths 190 and 175 mm, respectively (middle, Fig. 1g, 'Methods'). L1 was fixed 190 mm in front of TL to set up an IFP exactly at the TL. Next, L2 was positioned between L1 and TL to further increase laser convergence and thus shift the exact position of IFP away from the TL. Accordingly, IFP is always in front of the TL, with L2 determining its exact position: Simply shifting L2 along the laser path between 100 and 5 mm distance from the TL expanded the effective FOV diameter to anywhere between 1.2 and 1.8 mm, respectively (compare Fig. 1g, middle).

In nTC$_2$ (Fig. 1g, right), we replaced SL with a single lens (L3) of 200 mm focal length (LA1708, Thorlabs). L3 operated in much the same way as L2 in the previous modification M1, however now the IFP was behind rather than in front of TL. Depending on the position of L3, this yielded effective FOV diameters anywhere between 2.5 and 3.5 mm. Importantly, in each case effective image brightness remained approximately constant across the full FOV (Fig. S1c–f, Methods). Here, the marginal brightness increase towards the edges is related to the slight upwards bend in the imaging plane as commonly seen for large FOV 2P microscopes[5,8]—see also below. The axial difference between the edge and centre of the imaging plane was 20, 45, 87 and 170 μm for 1.2, 1.8, 2.5, 3.5 mm FOV, respectively.

Our design's full optical path and control logic are shown in Fig. 1h. All functions are executed from the scan software, which directly controls the xy-scan path as usual. To synchronize an electrically tunable lens (ETL, see below) and/or a Pockels cell to this xy-scan, a copy of the fast-mirror command is sent to two microcontrollers. Each of these then executes preloaded line-synchronized commands that are defined using a standalone graphical user interface (GUI). In this way, this standalone z-control-system only requires a copy of the scan mirror command, meaning that it can be directly added to a 2P microscope setup without the need for software modifications.

**Increased effective laser power.** Because our nTC design avoids overfilling of the objective's back aperture and uses fewer optical elements in the laser path (by replacing the native scan lens consisting of two doublets with one doublet and one singlet in nTC$_1$, or one singlet in nTC$_2$), total achievable laser power at the sample was increased approximately 4-fold compared to all configurations of the DL setup (Fig. 1j). This additional power could, for example, be used to facilitate imaging deep in the brain, or alternatively to drive additional setups from the same laser source. For instance, when imaging the small brains of larval zebrafish or fruit flies, there is rarely a need to exceed 50 mW, meaning that it is theoretically possible to drive ten such nTC setups from a single standard laser (e.g. Coherent Chameleon Vision-S Laser, average power ~1.5 W at 930–960 nm, assuming 50% loss through the setup).

**Spatial resolution under nTC.** To establish how our nTC approach affected the excitation PSF, we first imaged 175 nm fluorescent beads across all configurations at 927 nm wavelength and 15–20 mW laser power at the sample ('Methods', Supplementary discussion). Starting from a DL spot-volume of 0.56 and 3.15 μm (xy and z, respectively), our different modifications elongated and laterally expanded the PSF to varying degrees, from 0.77 (xy) and 9.94 (z) μm for the 1.2 mm FOV configuration to 2.21 (xy) and 41.49 (z) μm at 3.5 mm FOV (Fig. 1k–m). Accordingly, increasing the FOV using nTC mainly elongated the PSF, while restricting its lateral expansion. However, PSF expansions were generally stronger than in other large FOV 2P-approaches which, for example, reported ~15 μm at the edge of a 10 mm FOV[5] or <10 μm at the edge of a ~5 mm FOV[8], see also refs. [5,6]. These approaches achieve their optical results through custom made, large-diameter optics, which are generally more expensive and more difficult to retrofit to existing setups. Notwithstanding, the possibility of optically merging adjacent image structures strongly depends on the size and spatial distribution of labelled biological structures—a general limitation in optical microscopy, rather than a specific limitation to our nTC approach (discussed e.g. in ref. [26]).

The systematic effects on PSF shape across configurations also meant that our nTC approach could be used to flexibly match

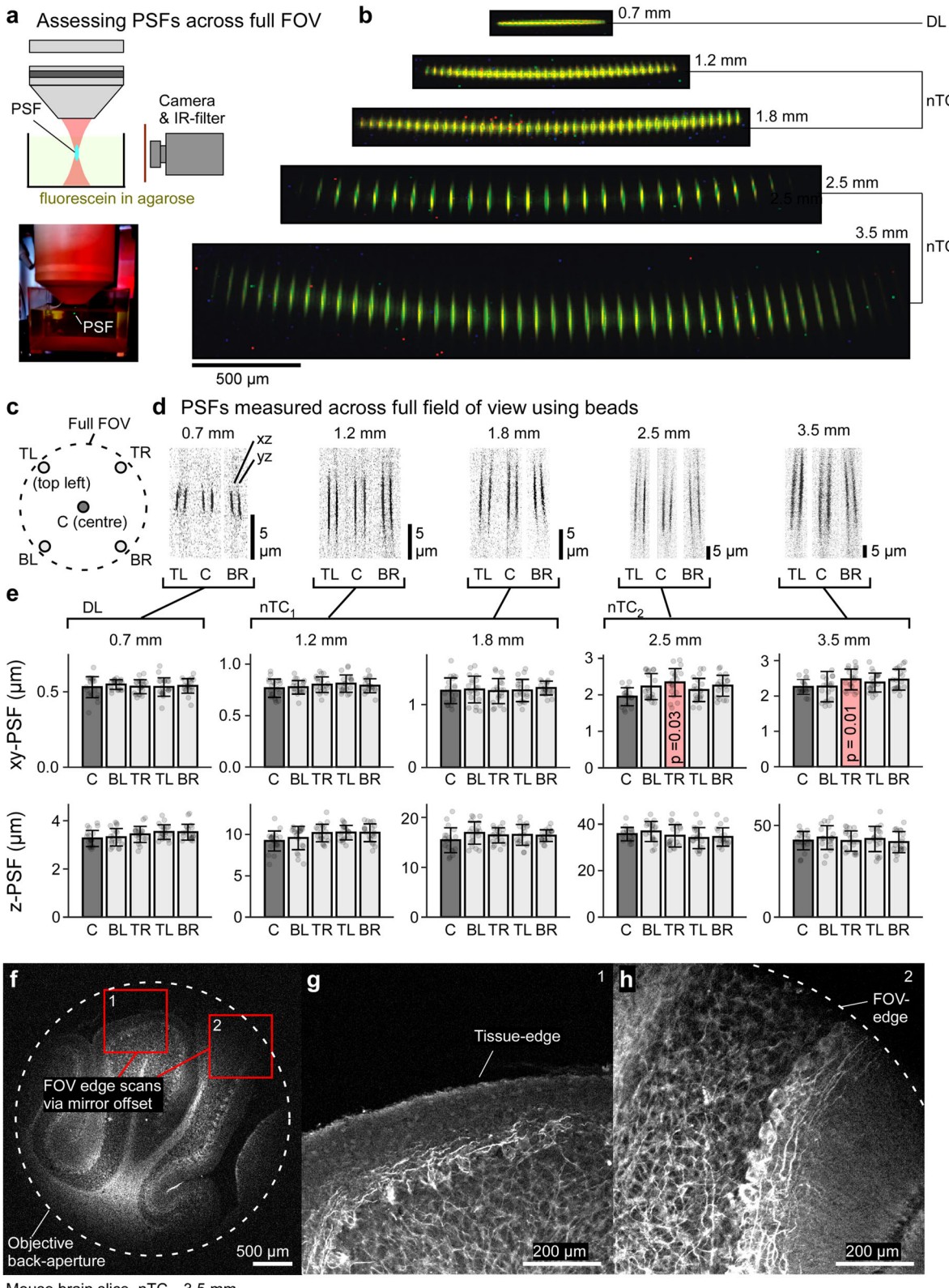

**f** Mouse brain slice, nTC₂, 3.5 mm

PSF dimensions to specific experimental needs. For example, the sub-micron DL PSF offered by typical collimated 2P-setups maximises spatial resolution which is invaluable for resolving small synaptic processes or the somata of larval fruit flies (typically < 5 μm). However, many species' cell bodies are much larger. For example, in the brain of larval zebrafish a very small DL PSF spatially typically oversamples the "mid-sized" ~5–10 μm somata at the expense of a potentially substantial loss in SNR. This limitation can be avoided by nTC-mediated adjustment of the PSF (cf. Figs. 2, 3). Similarly, for picking up somatic signals from cortical neurons in the mouse, a "10-fold expanded" ~5 μm PSF yields the best SNR[12].

**Fig. 2 Spatial resolution over the full field of view. a** Schematic (top) and photograph (bottom) of the setup used to directly film excitation volumes. IR, infrared. **b** effective scan-planes directly visualised as indicated in (**a**) for all optical configurations as indicated, in each case with scan-points spaced to facilitate inspection of individual PSFs. **c–e** PSF measurements using fluorescent beads (cf. Fig. 1m) across the full FOV for all optical configurations as indicated in (**c**). Shown are exemplary scan profiles (pairs of xz and yz projections) from (**d**) and (**e**) quantification of full width half maxima (FWHM) In lateral (xy) and axial dimensions (z). For statistical evaluation we compared each "lateral" PSF estimates to the respective centre estimates using 1-way ANOVA. No significant differences were detected, except in two top-right (TR) xy values for $nTC_2$ conditions, as indicated. These small differences are likely explained by slightly imperfect laser centring. Error bars in mean ± s.d., $n = 20$ experimentally independent samples per measurement. TL, top left, TR, top right, BL, bottom left, BR, bottom right. **f–h**, example scan under $nTC_2$ 3.5 mm configuration of a mouse brain slice labelled with SMI31 for phosphorylated Neurofilament-H (NFH). Full usable FOV (**f**) and two zoomed in regions (**g**, **h**) as indicated in (**f**), without moving the objective, to illustrate image quality at the FOV edges (**f–h**, 1024 × 1024 px, 0.49 Hz). Data leading to **e** in Source data file.

**Imaging across the full FOV**. To further assess how the different optical configurations impacted PSF-shapes across the whole FOV, we next visualised excitation volumes using a camera (Fig. 2a, b)[27]. Specifically, we positioned a fluorescein-solution-filled cuvette below the objective and filmed it from the side (Fig. 2a). Compared to imaging beads (Fig. 1k–m) this approach had the advantage that excitation volumes could be visualised much more directly, as well as across different positions in space in rapid succession (Supplementary Movie S2). However, the resultant scan profiles overestimated PSF sizes and were thus not suitable to determine their absolute dimensions – rather, the goal was to observe relative variations in PSF shapes, positions, and orientations over the full FOV. Figure 2b shows a direct, scale-matched visualisation of effective scan profiles for all optical configurations. This confirmed that the DL configuration had the smallest profiles, followed by increasing-FOV variations of $nTC_{1,2}$. Moreover, scan-profiles were curved to different degrees, with correspondingly tilted excitation volumes towards the edge[5,8]. If required, this can be part-corrected via the ETL. However, biological structures are rarely perfectly flat either. As described further below, often a more useful solution might be to instead fit the scan-plane curvature to the 3D curvature of the interrogated sample.

To quantify absolute PSF dimensions across the full FOV, we next returned to measuring fluorescent beads (cf. Fig. 1k), however, this time imaging both at the FOV centre (C) and at the scan edges (top left: TL, bottom right: BR, etc., cf. Fig. 2c) in each of the five optical configurations (Fig. 2d, e). This confirmed that while as expected PSFs slightly tilted outwards at the edges (Fig. 2d, cf. Fig. 2c), their traverse and axial dimensions did not change significantly (except in two cases where there was a slight xy-expansion as indicated, likely due to imperfect laser centring, Fig. 2e). Moreover, no noticeable image distortion across the FOV was detected, illustrated by imaging a fixed sample under nTC2 2.5 mm configuration at three different lateral translations, followed by direct superposition of the same selected structure in each image (Fig S2). Together, this implied that in practice, the full FOV was usable for imaging, as further confirmed in an $nTC_2$ 3.5 mm FOV example scan of a fixed mouse brain slice labelled with SMI31 for phosphorylated Neurofilament-H (NFH), staining the basket cells that wrap around Purkinje neurons (Fig. 2f–h). Here, thin neural processes could be followed essentially all the way to the FOV-edge, where it clipped due to the objective back aperture (e.g. Fig. 2h).

**Effective image resolution and brightness**. Next, we directly compared the different effective spatial resolutions and signal integration by imaging the same in vivo zebrafish sample in each configuration. For this, we sparsely expressed GCaMP6f under the pan-neuronal promotor HuC in larval zebrafish[28] and imaged one animal that randomly exhibited isolated and easily recognisable expression in neurons of the upper spinal cord, including one cell body (~7 μm diameter) and several individual synapses

(~1 μm diameter, arrowheads indicate matching position of 2 such synapses across scans, Fig. 3a). These image structures were consistently recognisable across all optical configurations. This notion was further confirmed in functional scans during visual stimulation that was time-interweaved with the scanner retrace to avoid crosstalk (Fig. 3b–d, 'Methods'). For example, full-field UV-flashes elicited different responses in different image structures on a trial-to-trial basis (e.g. see highlighted traces of $nTC_2$ condition). The different configurations also achieved notably different overall brightness in the imaged structures and allowed "optically merging" the spinal tract in larger FOV (and thus more elongated PSF) scans to different degrees (see also Supplementary Movie S3). Accordingly, the same set of neuronal elements could be usefully imaged across all optical configurations in this sample, allowing experimenter-controlled trade-off between FOV, 3D spatial resolution, and consequent signal integration.

**Rapid axial scans**. In addition to expanding the FOV, our de-collimated design also shifts the excitation point beyond the objective's nominal focal distance (Fig. S1b). The same optical effect can be exploited to drive rapid axial shifts in the excitation plane by the introduction of an electrically tunable lens (ETL) early in the laser path (Fig. 4, Fig. S3, c.f. Fig. 1h)[18,21]. Specifically, we positioned an off-the-shelf ETL (EL-16-40-TC-20D, Opto-tune) 200 mm in front of the first scan mirror and controlled it with a custom driver board (see user manual). In this position, already a minor deviation from the perfectly flat curvature at zero input current slightly converged the laser which, in turn, strongly shifted the effective z-focus below the objective. For example, in both $nTC_1$ and $nTC_2$, stepping the input current from zero to 25% (50 mA) gave rise to a ~600 μm z-shift of the excitation plane (Fig. 4a, b). The use of only a small fraction of the ETL's full dynamic range enabled short turnaround times (1–10 ms, depending on distance jumped, Fig. 4c, d, Fig. S3) and prevented overheating[18,29]. An overview of possible scan-paths is presented in Supplementary Movie S4. If required, rapid synchronization of the ETL curvature with a Pockels cell for controlling effective laser power at the sample plane can compensate for any systematic variations in image brightness associated with increased penetration depth.

Taken together, our design, therefore, presents a low-cost (~ £1000, cf. user manual) and easily implemented solution to expand the FOV of any 2P microscope in three dimensions. In the following, we demonstrate how these capabilities can be exploited in a range of neurophysiological applications in larval zebrafish, as well as the mouse cortex and fruit fly brain.

**Imaging zebrafish under 2P**. Owing to their small size and transparent larval stage, zebrafish have become a valuable model for interrogating brain-wide neural circuit function[30,31]. However, from tip to tail, the brain and spinal cord of a 7–9 *dpf* larval

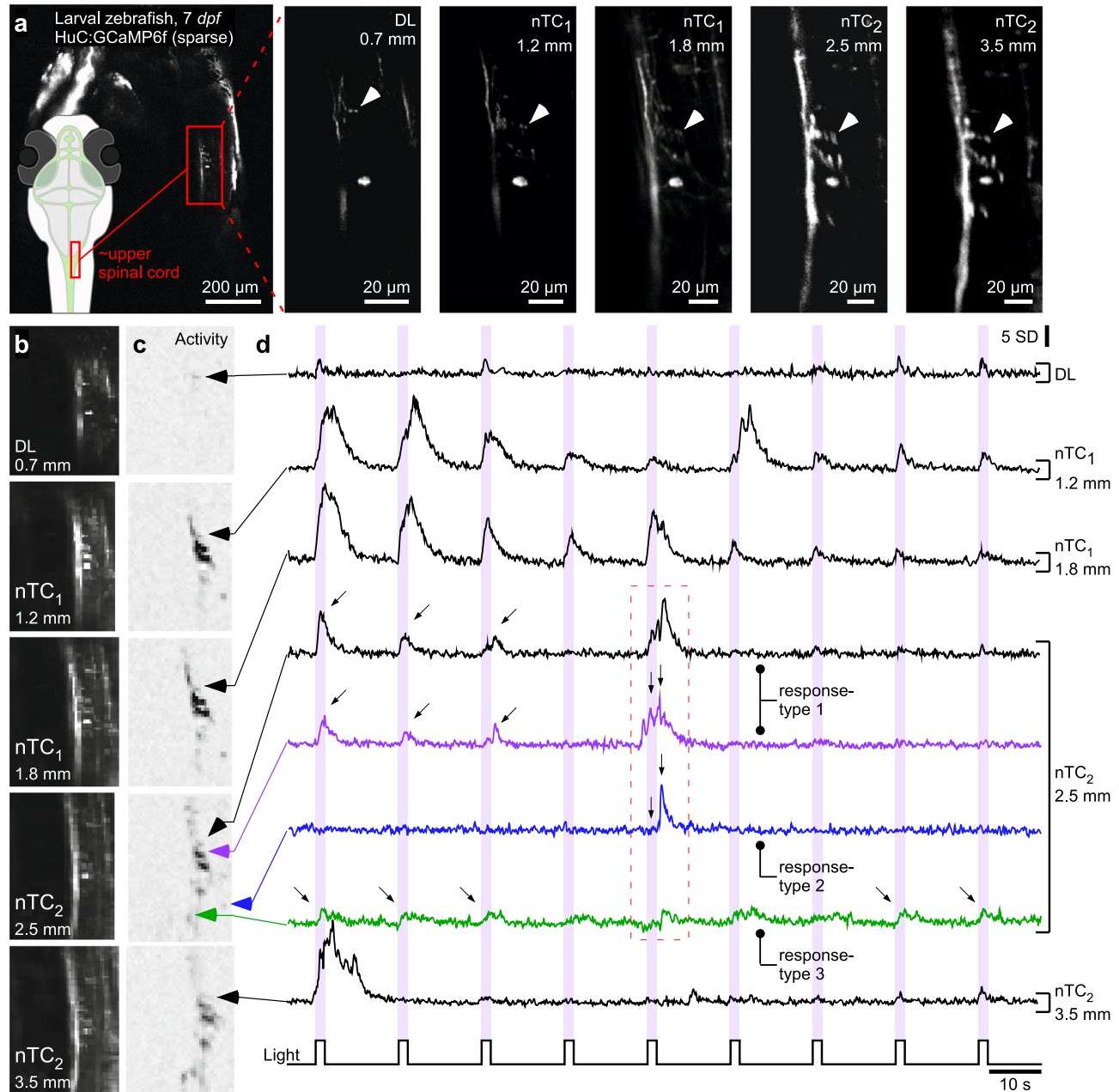

**Fig. 3 In vivo calcium imaging across different optical configurations. a** The same set of neurons of the 7 *dpf* larval zebrafish upper spinal cord (HuC:GCaMP6f, random sparse expression, see overview scan and schematic on the left) was imaged in all optical configurations as indicated at 512 × 512 px (1 Hz). Arrowheads highlight the same synaptic structures in each scan. **b–d** 64 × 64 px (7.81 Hz) activity scan from fields of view shown in (**a**) for all five configurations during presentation of full-field flashes of UV-light which stochastic elicited activity in these imaged neural structures. In each case the average scan projection (**b**) and neighbour-correlation based activity projection (**c**) are shown (hereafter referred to as "activity-correlation"). Darker shadings, equalised for visibility, denotes increased local activity (for details, see ref. [70]). Black traces in (**d**) show time-traces for the same structure in all cases. For the nTC₂ 2.5 mm FOV condition, time-traces from different neural structures are extracted to illustrate different responses in different structures. All activity traces in this and the following figures are shown in z-scores relative to their own baseline (hence y-scale in s.d.). We choose this metric over dF/F as it emphasises detectability of events rather than the relative change from the indicator's baseline fluorescence, which differs between biosensors.

zebrafish reaches about 3.5–4.5 mm, with the central brain occupying approximately 1.2 mm in length and 0.7 mm in width. This is too large to fit into the FOV of a typical DL 2P setup. As a consequence, studies routinely "tile-scan" the brain in sequential stages to provide brain-wide data[32,33].

On the other hand, the transparent body wall of larval zebrafish makes them well-suited for 1-photon selective-plane-illumination microscopy (1p-SPIM / "lightsheet microscopy"), which is not

FOV-limited in the same way as 2P microscopy[29,30,34]. However, 1p-SPIM and related techniques[35] have a number of drawbacks, including constraints on achieving a homogenous image due to scattering and divergence of the excitation light with increasing lateral depth[2], limited access to tissues that are shadowed by strongly-scattering tissue such as the eyes[36,37] and, critically, a direct and bidirectional interference between the imaging system itself and any light stimuli applied for studying zebrafish vision[38].

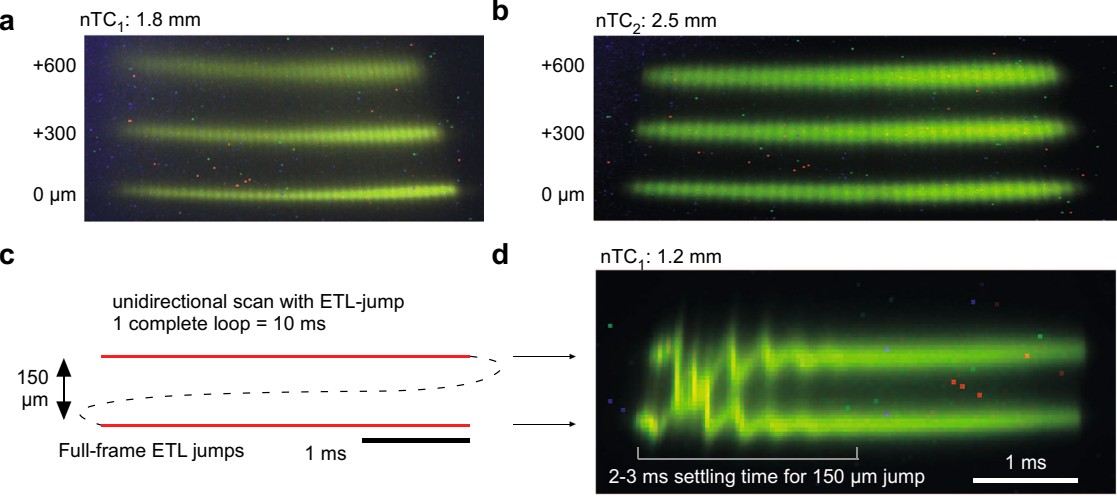

**Fig. 4 Rapid remote focussing. a, b**, Scan-profiles with the electrically tunable lens (ETL) "flat" (zero input current, lowest profile) and engaged to achieved axially elevated scan planes at +300 and +600 μm (middle, upper profiles, respectively) in nTC$_1$ (**a**) and nTC$_2$ (**b**) configuration, as indicated. Associated size-changes in the effective full field of view were generally <5% (compare top and bottom planes). In each case, axial-shifts required <25% unidirectional peak current on the ETL which in turn facilitated rapid ETL-settling times: **c, d**, Schematic (**c**) and measured (**d**) axial jumps and settling time: the ETL was programmed to iteratively focus up and down by 150 μm at each end of two long (5 ms) scan lines, as indicated. This enabled a direct read-out of ETL settling at each line-onset (oscillations in **d**). For the 150 μm jumps shown, oscillations decayed below detectability within 2–3 ms. For corresponding readouts of the ETL-position signal, see Fig. S3.

These specific challenges could be readily addressed by our nTC 2P setup. To demonstrate this, we imaged larval zebrafish in a range of optical configurations.

**Mesoscale whole-zebrafish 2P imaging**. First, we used the 3.5 mm configuration of nTC$_2$ to capture the largest-possible FOV of two larval zebrafish facing each other. This configuration comfortably allowed simultaneous mesoscale imaging of two entire zebrafish brains, here responding to full-field flashes of UV-light (Fig. 5a–e, Supplementary Movie S5). Alternatively, the same configuration could be used to capture the entire central nervous system of one fish in a single frame, including the brain and nearly up to the tip of the spinal cord (Fig. 5f). Zooming in throughout the sample enabled resolving cellular details (Fig. 5g–i).

Next, beyond mesoscale imaging, many studies of zebrafish neuronal function focus on either the brain or the spinal cord (rather than both). In this case, using the more highly resolved nTC$_1$ configuration with 1.2 mm FOV may be preferable; this just about fits one full zebrafish brain at a time.

**3D random access scanning across the zebrafish eye and brain**. In the nervous system, key functionally linked circuits are often separated in 3D space, representing a general problem for systems neuroscience. For example, the retinal ganglion cells of the zebrafish eye project to the contralateral tectum and pretectum, which are both axially and laterally displaced by several 100 s of microns. Accordingly, it has been difficult to simultaneously record at both sites, for example, to study how the output of the eye is linked to the visual input to the brain. To address this problem, we used our nTC$_1$ configuration in synchronisation with the ETL to establish quasi-simultaneous 3D random access scanning of the zebrafish's retinal ganglion cells across both the eye and brain (Fig. 6a–c). For this we used an Islet2b:mGCaMP6f line which labels the majority of retinal ganglion cells in larval zebrafish. We first defined a slow, high-spatial resolution scan (512 × 512 px, 0.98 Hz) that captured the entire front of the head, however with a single z-jump at the centre of the frame to set-up a "staircase-shaped" scan-path (Fig. 6b, c). Here, empirical adjustment of the magnitude of the z-jump allowed us to identify the axonal processes of retinal ganglion cells

in the brain, and their dendritic processes in the contralateral eye in the top and bottom of the same imaging frame, respectively. Based on this image, we next defined two scan regions for 3D random access scanning, one capturing a single plane across the tectum, while the other captured a smaller area of a subset of RGC dendrites and somata in the eye (Fig. 6d–f). Finally, we decreased the spatial resolution to 64 × 64 px to quasi-simultaneously image both regions at 7.81 Hz. This configuration allowed reliable recording of light-driven signals from individual RGC neurites across the eye and brain (Fig. 6g–j). Next, we repeated this experiment, however, this time in zebrafish larvae that were transiently injected with Islet2b:mGCaMP6f plasmid. These animals stochastically express mGCaMP6f in only a very small number of RGCs, making it possible in principle to identify the processes belonging to the same RGC in both the eye and brain. As a proof of principle, we present one such experiment where we could clearly image the processes of single RGCs at both sites (Fig. 6k–o). For this type of application, it will be important to optimise the genetic protocol to improve expression levels and thereby facilitate the identification of the same RGC's processes at both sites.

**3D plane-bending for imaging activity across the zebrafish brain**. During standard planar scans of the larval zebrafish brain, the powerful optical sectioning afforded by the 2P approach highlights the 3D curvature of distinct brain regions by cutting right across them (Fig. 7a–d). While it was possible to quasi-simultaneously image anywhere within the brain at high spatial resolution using nTC$_1$, a planar scan grossly misrepresented the real 3D structure of the zebrafish brain (Fig. 7d, top panel). For example, the tectum in larval zebrafish is tilted upwards ~30°[39], meaning that rather than either cleanly sampling across its reti-notopically organized surface, or perpendicularly across its stacked functional layers, the planar image instead cut the tectum at an effective 30° angle to yield a mixture of both, thus con-founding interpretation. To ameliorate these issues, we used a 3D curved scan plane by driving the ETL as a sqrt(cosine) function of the slow y-mirror command ('Methods'). This enabled z-curvature "halfpipe" scans that could be empirically fitted to follow the natural curvature of the brain, thereby closely

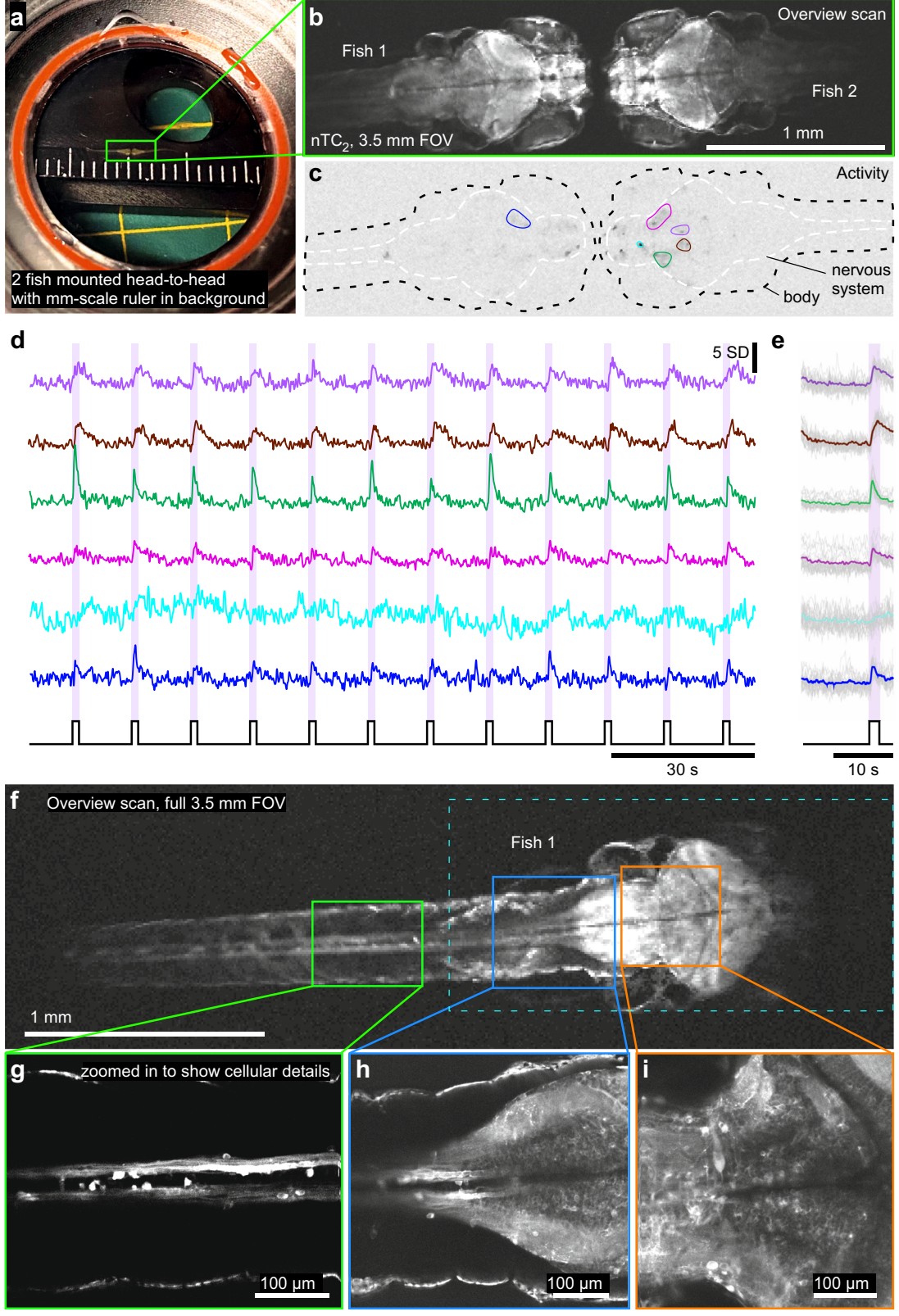

**Fig. 5 Mesoscale imaging of zebrafish larvae. a** Photograph of two 9 *dpf* zebrafish larvae mounted head-to-head in a microscope chamber with mm-scale ruler in background. **b** The same 2 fish (HuC:GCaMP6f) as in (**a**) imaged under 2-photon with nTC₂ 3.5 mm FOV configuration, at 512 × 128 px (3.91 Hz). **c**, **d** Activity-correlation (cf. Fig. 3c) of the scan in (**b**) during presentation of full-field flashes of ultraviolet (UV)-light, with hand-selected exemplary ROIs, extracted time-traces (**d**) and light-stimulus-aligned averages (**e**). **f-i** The same fish as shown on the left in (**b**, fish 1), now shown at full 3.5 mm field of view (f, 512 × 128 px, 3.91 Hz) and increased spatial resolution scans of regions as indicated to reveal cellular detail (**g-i**, 1024 × 1024 px, 0.49 Hz). RGC, retinal ganglion cell.

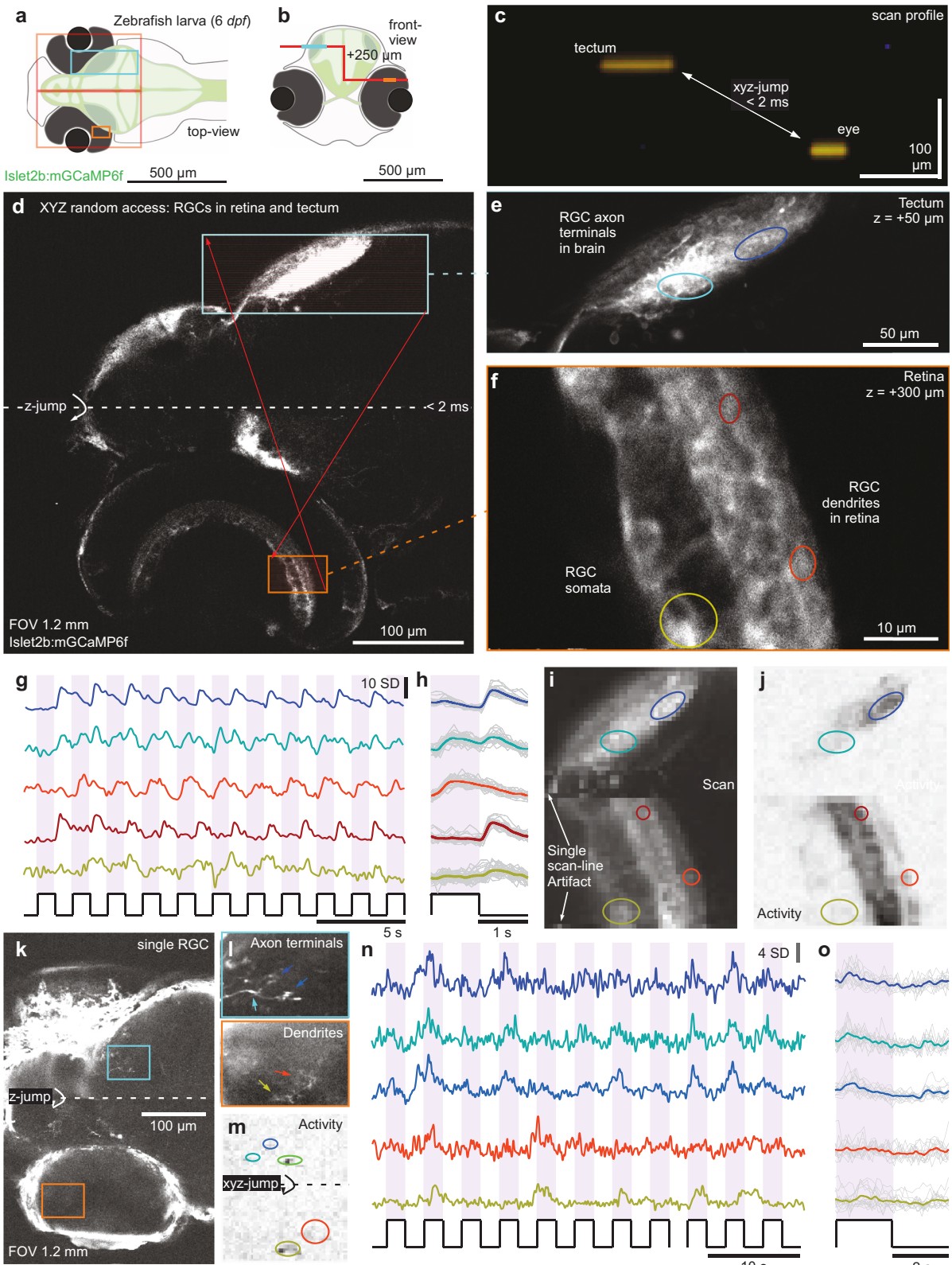

**Fig. 6 3D random access scanning of the zebrafish eye and brain. a**, **b** Schematic of zebrafish larva from top (**a**) and front (**b**) with scan configurations indicated. **c** direct x-z visualisation of the scan-profile used in the below. **d** nTC₁ 1024 × 1024 px scan across an Islet2b:mGCaMP6f 6 *dpf* larval zebrafish eye and brain. At the centre of the scan, the axial focus is shifted upwards such that the axonal processes of retinal ganglion cells (RGCs) in the tectum (top) and their somata and dendritic processes in the eye (bottom) can be quasi-simultaneously captured. **e**, **f** 1024 × 1024 px split-plane random access jump between tectum (**e**) and eye (**f**) and **g**–**j** 2 times 64 × 128 px (15.6 Hz) random access scan of the same scan regions with raw (**g**) and event-averaged (**h**) fluorescence traces, mean image (**i**) and activity-correlation (**j**, cf. Fig. 3c). The stimulus was a series of full-field broadband flashes of light as indicated. **k**–**o** as (**d**–**j**), with individual RGCs transiently expressing GCaMP6f under the same promoter.

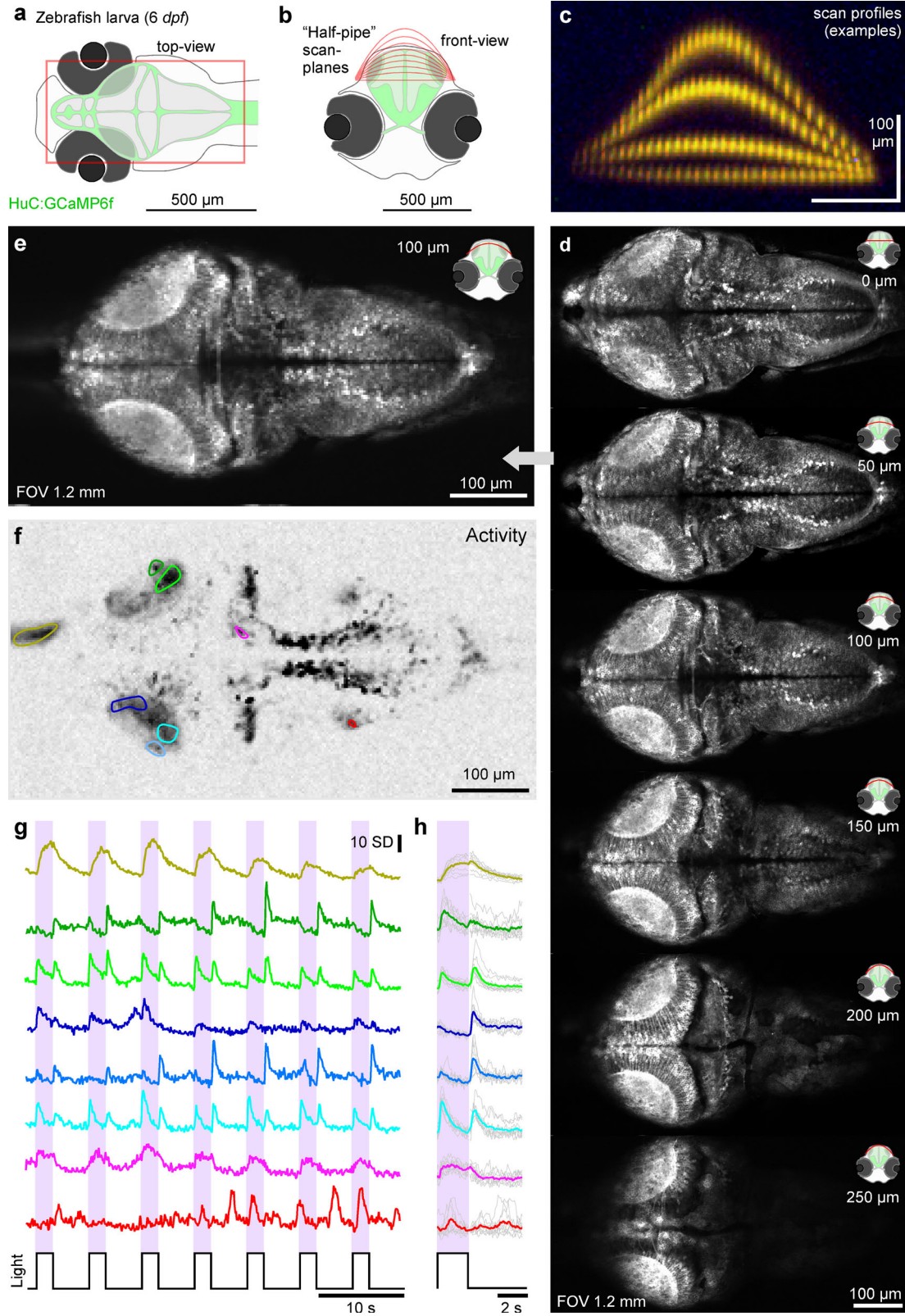

**Fig. 7 2P plane-bending to image the in vivo larval zebrafish brain. a–c** Schematic of HuC:GCaMP6f larval zebrafish brain viewed from top (**a**) and front (**b**) with scan planes indicated, and (**c**) example-scan-profiles. **d** nTC$_1$ 512 × 1024 scans of a 6 *dpf* zebrafish brain with different plane curvatures, with peak axial displacement at scan centre as indicated. At curvatures ~100–150 µm peak displacement the scan approximately traverses the surface of the tectum. **e–h** Mean (**e**), activity-correlation (**f**, cf. Fig. 3c) and fluorescence traces (**g**, raw and **h**, event-triggered mean) from a 170 × 340 px scan (5.88 Hz) of the 100 µm peak displacement configuration (image 3 in (**d**)). The fish was presented with full-field and spectrally broad (~360–650 nm) series of light-flashes. See also Fig. S4.

capturing the functional anatomical organisation of the zebrafish brain (Fig. 7b–d, Supplementary Movie S6). From here, we chose a single halfpipe plane that best followed the curvature of the two tecta and imaged this plane at 7.81 Hz (256 × 128 px, 1 ms/line, Fig. 7e). We then presented spectrally broad full-field light stimulation. This allowed us to interrogate brain-wide visual function in response to arbitrary wavelength light (Fig. 7f–h). As required, the halfpipe scans could also be staggered for multiplane imaging at correspondingly lower image rates, including negative bends that surveyed the difficult-to-reach bottom of the brain between the eyes (Fig. S4, Supplementary Movie S7).

**Mesoscale and 3D random access imaging of the mouse brain.**
The width of the adult mouse's brain is ~10 mm[40] which makes it too large to be comprehensively captured by conventional 2 P microscopy. Here, an experimental goal might be to reliably resolve the ~20+ μm somata of major cortical or subcortical neurons across a 10 mm FOV. At the Nyquist detection limit, this would "only" require ~1000 pixels across, which is well within the range of standard high-resolution scan-configurations. Accordingly, currently the main limitation in achieving this goal is the microscope's maximal FOV. Our nTC design makes important steps to address this limitation.

When configured for a 3.5 mm FOV (nTC$_2$), our setup captures about a third of the width of a mouse's brain. In this configuration, a scan of a transverse section from a Thy1:G-CaMP6f mouse (Fig. 8a, b) illustrates how the objective's back aperture casts a shadow at the image edge, thus limiting the spatial extent of the scan (Fig. 8c). Within this maximal window, a high-resolution 1024 × 1024 px scan allowed us to resolve the somata of major cortical and hippocampal neurons (Fig. 8d, Supplementary Movie S8). Accordingly, at this largest FOV configuration, effective signal detection largely sufficed to capture the mouse brain's major neuron populations. However, with our galvo-galvo setup, scan rates at this level of spatial detail were slow (0.49 Hz, 2 ms/line). Accordingly, we used a mesoscale imaging approach with reduced spatial sampling (256 × 256 px, 1 ms/line) to capture the entire image at 3.91 Hz. This permitted simultaneous population-level "brain-wide" recording of seizure-like activity across the cortex and underlying hippocampus following bath application of an epileptogenic (high K$^+$, zero Mg$^{2+}$) solution (Fig. 8e–g). To demonstrate the value of the system for more detailed readout of neuronal activity, we also used random access scans to simultaneously capture distant smaller scan-fields at high resolution, both spatially and temporally (two times 256 × 128 px at 3.91 Hz, Fig. 8d, h–l, Supplementary Movie S8). In the example provided, the laser travelled between the two scan fields separated by ~1 mm within two 1 ms scan lines. This allowed us to record quasi-simultaneous neural activity across both the cortex and hippocampus. The generally high SNR in these recordings also suggested that additional temporal or spatial resolution could be gained by the use of resonance scanners in place of our galvos[41].

The large FOV nTC$_2$ configuration also lends itself to imaging mouse cortical neurons activity in vivo (Fig. 9), an increasingly common demand in neuroscience. Here, the maximal 3.5 mm FOV captured an entire cranial window of a Thy1-GCaMP6f mouse prepared for optical interrogation of the somatosensory cortex, comprising an estimated 10,000+ neurons in a given image plane (Fig. 9a–c, Supplementary Movie S9). Even in an intermediate nTC$_1$ configuration (in this case a 1.5 mm FOV) the full image still comprised several 1000 s of neurons (Fig. 9d), many more than could be simultaneously captured at scan-rates suitable for functional circuit interrogation with a galvo-galvo setup. In an example scan we again used a random-access

approach to quasi-simultaneously record two 330 × 210 μm regions separated by ~1.2 mm (two times 128 × 64 px at 7.81 Hz). As in the brain slice preparation (Fig. 8), this reliably resolved individual neurons in spatially distinct regions of the mouse brain (Fig. 9d–i). Finally, we also recruited the ETL to set up an axially tilted scan plane. This allowed quasi-simultaneous recording from neurons separated several hundreds of μm in depth across layers 1–4 of the mouse cortex in vivo (Fig. 9j–l).

**Multi-plane circuit mapping with optogenetics in *Drosophila*.**
Despite the generally enlarged FOV and concomitant increase in the PSF, our setup was still capable of resolving details of small neural processes in the <0.1 mm diameter nervous system of a first instar larval fruit fly. To assess the difference in image-resolution between our nTC setup and a DL-configuration, we first obtained anatomical scans from a third instar VGlut:GCaMP6f larva which expressed GCaMP6f in structurally well-defined neurons of the ventral nerve cord (Fig. 10a, b). This revealed that while the DL image was clearly sharper (Fig. 10a), the nTC$_1$ system nevertheless comfortably delineated individual somata (Fig. 10b).

*Drosophila* was an ideal preparation to demonstrate our system's capacity for multi-plane imaging for optogenetic functional circuit mapping (Fig. 10c–k). At the first larval stage, the height of the brain excluding the ventral nerve cord is in the order of ~60–70 μm. Assuming an axial capture of ~3 μm per plane in a DL configuration (cf. Fig. 1o), comprehensively sampling from the whole brain would therefore require upwards of 20 planes (Fig. 10c). Here, the slightly elongated PSF of the nTC 1.2 mm configuration served as a useful compromise between spatial resolution and sampling density (Fig. 10d). To demonstrate the sampling that can be achieved under these conditions, we used a transgenic first instar larva that expressed the red-shifted optogenetic effector CsChrimson in all olfactory-sensory neurons (OSNs) on a background of pan-neuronal GCaMP6s (elav:GCaMP6s) (Fig. 10e, f). To reveal any potential bilateral crosstalk of olfactory signal processing across the brain's two hemispheres, one of the olfactory nerves was cut. We set up six image planes (six times 340 × 170 px), each separated by ~15 μm which together captured the entire brain across both hemispheres at ~1 Hz (Fig. 10d, f). In this configuration, presentation of 2 s flashes of red light from a scanline-synchronized 590 nm LED activated olfactory sensory neurons (OSNs). These in turn propagated the signal to higher processing centres, which we visualised as regionally restricted GCaMP6s responses in the brain (Fig. 10g–k, Supplementary Movie S10). The most strongly activated region was the ipsilateral antennal lobe (AL) (see also Fig. S5) which is directly innervated by the still-intact OSNs. Similarly, the olfactory second order processing centres, the mushroom body and the lateral horn, showed clear ipsilateral activation. In addition to these three major olfactory centres and their connecting tracts (e.g. plane 3), further processes and somata across both the ipsi- and contralateral lobe were also activated. Taken together, despite the slight expansion of the DL excitation spot, our nTC setup nevertheless allowed us to delineate key structural and functional information in this small insect brain.

## Discussion

The ongoing development of sophisticated optical probes to report on key biophysical events has increasingly raised the demand in neuroscience for high SNR and large FOV 2P microscopes. To date, however, these characteristics are almost exclusively limited to high-end and, inevitably, high-cost platforms. Here, we exploit the fact that in 2P microscopy there is no "traditional" collection plane, allowing us to deviate from the diffraction limited regime that is typically used in systems where the planes of excitation and collection must superimpose to avoid image blur. Instead, we propose a simple core modification of the

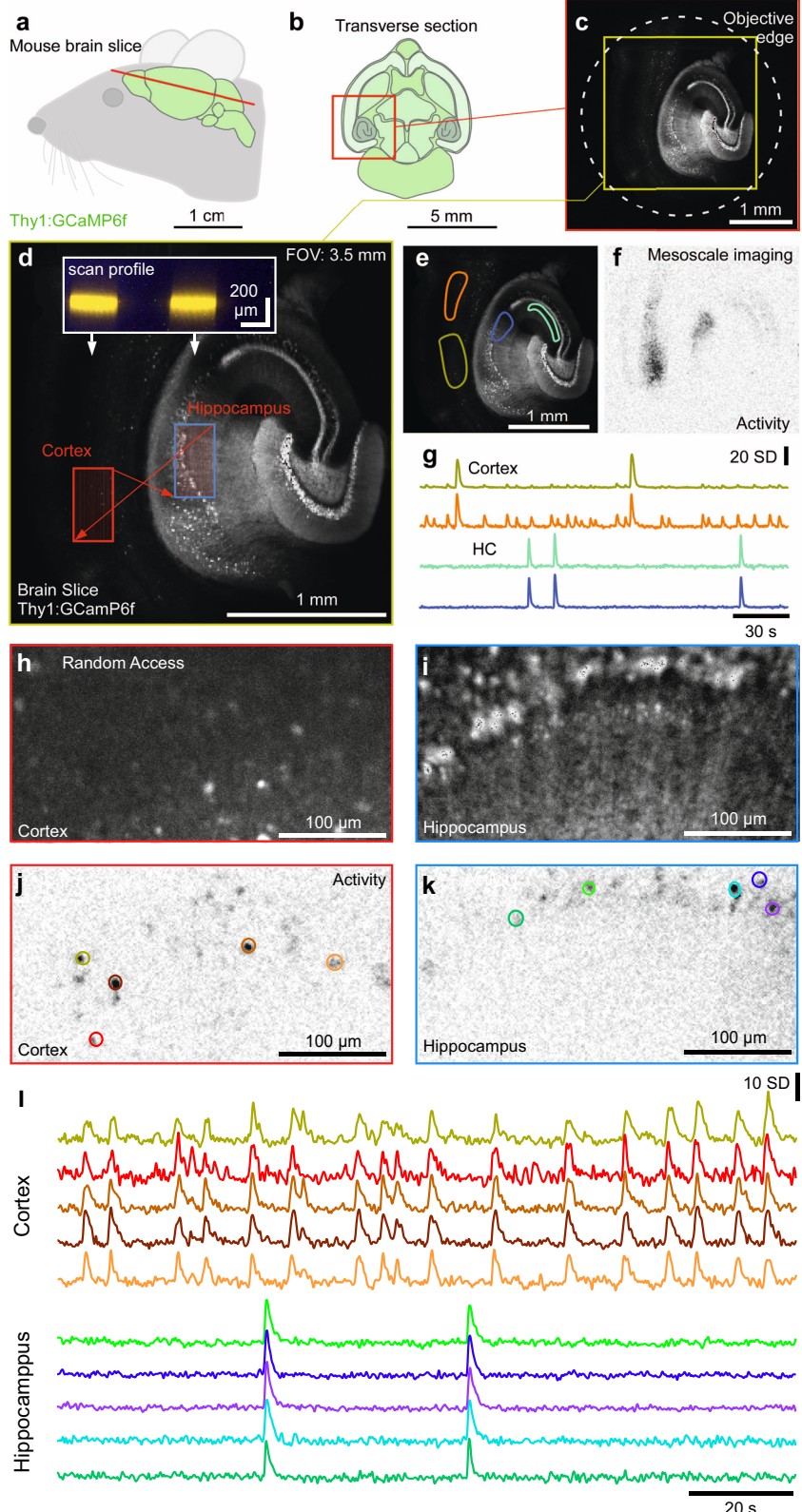

laser path that allows upgrading an out-of-the-box Sutter MOM DL 2P microscope into a system capable of performing high SNR and large-FOV volumetric scans. We demonstrate the capabilities of this system for interrogating dynamic events in the brains of a range of key model species that are already widely used in

neuroscience research. Since the core modification only requires the user to swap the scan lens for one or two off-the-shelf lenses, it can be tested (and fully reversed) within a matter of hours without the need for optical re-alignment or calibration. We anticipate that the simplicity and cost-effectiveness of this

**Fig. 8 Mesoscale and random-access imaging in mouse brain slice. a, b** Schematic of brain (**a**) and transverse section (**b**) of a Thy1:GCaMP6f mouse. **c, d** 1024 × 1024 px $nTC_2$ example scan of slice through cortex and hippocampus at maximal FOV (**c**) and $nTC_2$ zoom in (**d**) as indicated. Red arrows indicate rapid transitions between scan regions, with the inset showing scan-profiles. The slice was bathed in an epileptogenic (high K[+], zero Mg[2+]) solution to elicit seizures. **e–g** Mean of 256 × 256 px scan (3.91 Hz) of (**d**) with regions of interest (ROIs) indicated (**e**), activity-correlation projection (Methods) indicating regions within the scan showing regions of activity computed as mean correlation of each pixel's activity over time to all its neighbours (for details, see ref. [70]) (**f**) and z-normalised fluorescence traces (**g**). **h–l**, 2 times 128 × 256 px (3.91 Hz) random access scan of two regions as indicated in (**d**) allows quasi-simultaneous imaging of the cortex (**h**) and hippocampus (**i**) at increased spatial resolution, with activity-correlation (**j, k**, cf. Fig. 3c) and fluorescence traces (**l**) extracted as in (**j, k**).

solution and the significant enhancement in 2P imaging capabilities that it permits, will lead to its wide adoption by the neuroscience community.

**Combining an nTC approach with existing custom 2P designs.** The estimation of metrics that meaningfully compare the capability of our nTC design with other custom solutions is difficult, as these can depend strongly on the specific objective (N.A., back aperture size, working distance (focus)), its distance from the tube lens, and indeed the nature of the interrogated sample and the biological question itself. Rather, because our nTC approach fundamentally differs from traditional DL optics, it opens the possibility to further enhance the capabilities of existing custom 2P microscope designs.

A key benefit of our nTC approach is the flexibility that it offers. It can be seamlessly implemented on setups with galvanometric or resonant mirrors to work with a wide range of scan-strategies. Here, the "extra" optical magnification afforded by the FOV expansion means that scan-mirror and ETL movements translate to relatively larger xy or z-translations, respectively, making it easy to rapidly execute complex and large-scale 3D scan-paths. Our approach could principally also be combined with existing setups that use rapid piezo-positioning of the objective for axial scans, although in this case the objective movement relative to the tube lens will generate small but systematic variations in FOV and PSF shape. Accordingly, the use of remote focussing before the scan-mirrors is likely to be preferable in most applications.

Like in most 2P designs, our use of a Gaussian beam does not permit the generation of a truly arbitrary PSF shape. Nevertheless, if used in combination with temporal focussing[12,42] it would, in principle, be possible to modulate axial PSF expansion without strongly affecting lateral expansion, thus facilitating a greater range of PSF shapes. Similarly, an optimized design of the objective lens[43] and other optical elements[4] including the use of large diameter lenses to minimize aberrations[5], could all be combined with our optical design to further enhance the quality of 2P excitation.

**nTC and optical aberrations.** In general, beyond the PSF expansion that results from bypassing the objective's infinity correction (Figs. 1, 2, Fig. S1), and the increased field curvature that inevitably follows from FOV expansion, the change from a standard 2P DL-setup to an nTC configuration does not notably worsen other optical aberrations. In short, chromatic aberrations (which necessitate complex optical corrections in 1P microscopy) do not apply in 2P microscopy, because the excitation laser is essentially monochromatic and collection is spatially invariant. Instead, spherical aberrations tend to be dominant in 2P microscopy, i.e. when peripheral and axial rays do not converge to a point[44–49]. Here, the optical element that has the largest impact is the objective, which is not changed under nTC. Further monochromatic aberrations are mainly related to the sample structure and surrounding (immersion) medium itself. In the future, it will

be useful to explore how adaptive optics can address many of the above points, including spherical aberrations as well as coma and astigmatism[46]—see also Supplementary Discussion.

**Axial signal integration.** Beyond matching the PSF to a given biological application (by correspondingly switching between nTC configurations), the use of a non-DL excitation spot can also bring about additional benefits. First, the lower effective excitation N.A. produces a narrower light cone which is less likely to be scattered by tissue inhomogeneities[50]. Second, objects that are smaller than the focal excitation volume become dimmer, while objects that are similar in size or larger remain bright[8,51]. Third, PSF expansion also reduces photobleaching and photodamage which can have a more-than-quadratic intensity dependence[52,53]. For example, when using the large PSF of the 3.5 mm FOV configuration, it was possible to use up to 250 mW laser power without causing notable damage when imaging deep in the mouse cortex[54]. Here, calculations and experimental experience suggest that in general, our strategy of underfilling the objective's back aperture will greatly ameliorate photodamage[48,52,53]. Notwithstanding, any axial expansion in the PSF must be balanced with potentially undesirable merging of distinct image structures separated in depth.

Taken together, our nTC approach offers key advantages over traditional DL 2P microscopy, including the capacity for an increased FOV, rapid z-travel through minimal ETL commands, and overall increased laser power at the sample plane[48,55,56]. Moreover, it could principally be combined with a wide range of existing customisations to further push the capabilities of 2P microscopy in general. At the same time, our nTC approach is cost effective and can be readily implemented with minimal need for optical alignments and calibration.

## Methods

**Animal experiments.** All animal experiments presented in this work were carried out in accordance with the UK Animal (Scientific Procedures) Act 1986 and institutional regulations at the University of Sussex. All procedures were carried out in accordance with institutional, national (UK Home Office PPL70/8400 (mice), PPL/PE08A2AD2 (zebrafish)) and international (EU directive 2010/63/EU) regulations for the care and use of animals in research.

**Zebrafish larvae preparation and in-vivo imaging.** Zebrafish were housed under a standard 14:10 day/night rhythm and fed 3 times a day. Animals were grown in 200 mM 1-phenyl-2-thiourea (Sigma) from 1 day post fertilization (dpf) to prevent melanogenesis. Preparation and mounting of zebrafish larvae was carried out as described previously[57]. In brief, we used 6–7 dpf zebrafish (Danio rerio) larvae that were immobilised in 2% low melting point agarose (Fisher Scientific, Cat: BP1360-100), placed on the side on a glass coverslip and submerged in fish water. For eye-brain imaging, eye movements were prevented by injection of a-bungarotoxin (1 nL of 2 mg/ml; Tocris, Cat: 2133) into the ocular muscles behind the eye. Transgenic lines used were Islet2b:mGCaMP6f (eye-brain imaging) and HuC:GCaMP6f[28] (image of 3 zebrafish in same FOV). Zebrafish were imaged at 930 nm and 30–60 or 50–100 mW for brain and eye imaging, respectively.

**Creation of Islet2b:mGCaMP6f transgenic line.** Tg(isl2b:nlsTrpR, tUAS:memG-CaMP6f) was generated by co-injecting pTol2-isl2b-hlsTrpR-pA and pBH-tUAS-memGaMP6f-pA plasmids into single-cell stage eggs. Injected fish were out-crossed with wild-type fish to screen for founders. Positive progenies were raised to

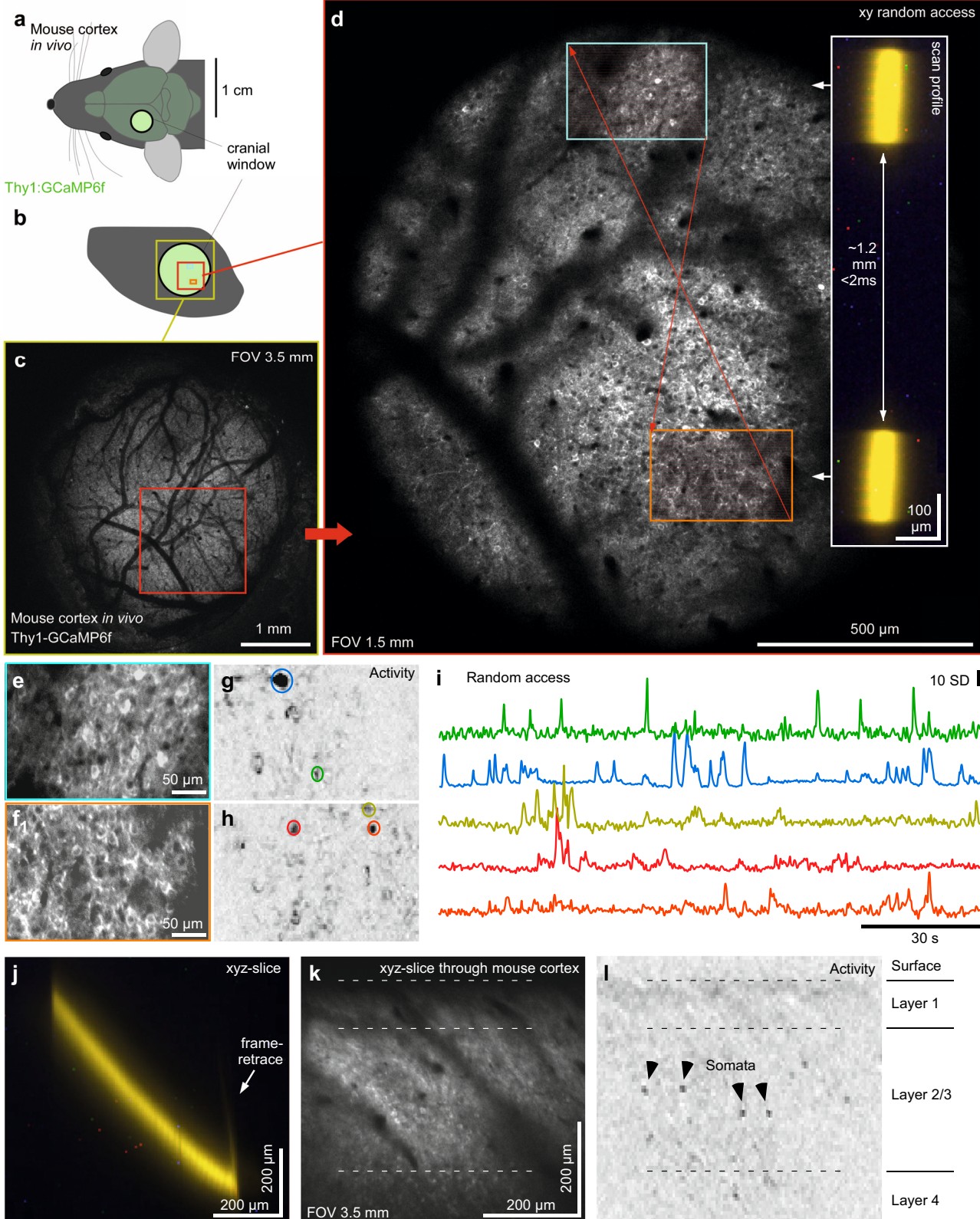

**Fig. 9 Mesoscale random-access imaging of mouse cortex in vivo. a,b** Schematic of Thy1:GCaMP6f mouse brain in vivo (**a**) with cranial window over the somatosensory cortex (**b**). **c**, **d** 1024 × 1024 px nTC$_2$ (**c**) and nTC$_1$ (**d**) images as indicated. Red arrows indicate rapid transitions between scan regions, with the inset indicating the scan-profile. **e–i** 2 times 128 × 256 px (3.91 Hz) random access scan as indicated in (**d**) with mean-projection (**e**, **f**), activity-correlation (**g**, **h**, cf. Fig. 3c) and fluorescence traces (**i**), taken from the ROIs as indicated in (**g**, **h**). **j–l** nTC$_1$ 128 × 128 px xyz-tilted plane (7.82 Hz) traversing through cortical layers 1–4 at ~45° relative to vertical (such that the x-image dimension corresponds to the x-mirror, while the y-dimension in the image represents simultaneous and matched y- and z-movement, with mean image (**k**) and activity-correlation (**i**, cf. Fig. 3c). The scan was taken under nTC$_2$ 3.5 mm FOV configuration and zoomed in to the central ~0.6 mm.

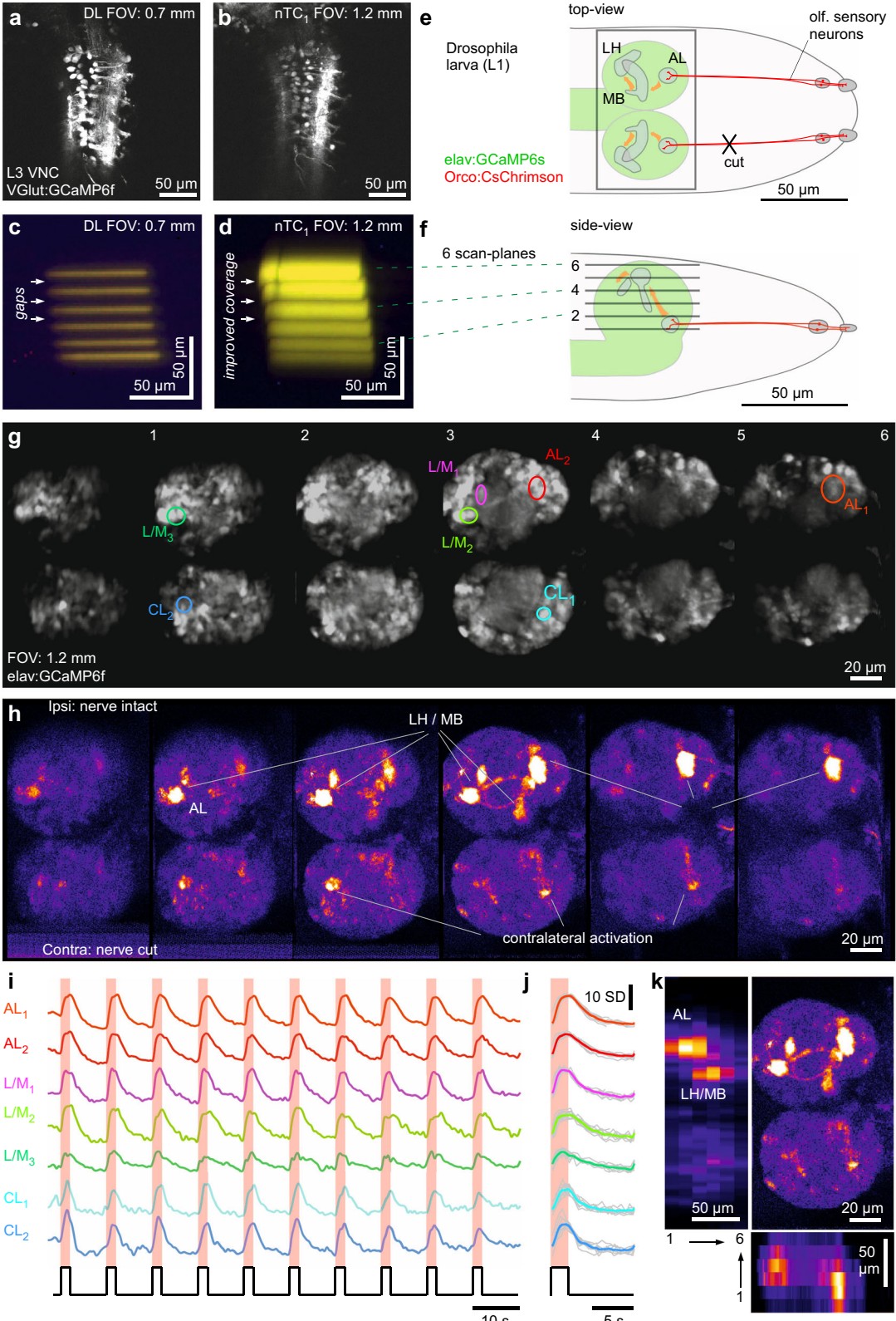

establish transgenic lines. All plasmids were made using the Gateway system (ThermoFisher, 12538120) with combinations of entry and destination plasmids as follows: pTol2-isl2b-nlsTrpR-pA: pTol2pA[58], p5E-isl2b[59], pME-nlsTrpR[60], p3E-pA[58]; pBH-tUAS-memGaMP6f-pA: pBH[61], p5E-tUAS[60], pME-memGCaMP6f, p3E-pA. Plasmid pME-memGCaMP6f was generated by inserting a polymerase chain reaction (PCR)-amplified membrane targeting sequence from GAP-43[62] into pME plasmid and subsequently inserting a PCR amplified GCaMP6f[63] at the 3′ end of the membrane targeting sequence.

**Acute brain slices**. One to two-month-old male Thy1-GCaMP6f-GP5.17[64] mice were used. Acute transverse brain slices (300 µm) were prepared using a vibroslicer (VT1200S, Leica Microsystems, Germany) in ice-cold artificial cerebrospinal fluid (ACSF) containing (in mM): 125 NaCl, 2.5 KCl, 25 glucose, 1.25 $NaH_2PO_4$, 26 $NaHCO_3$, 1 $MgCl_2$, 2 $CaCl_2$ (bubbled with 95% $O_2$ and 5% $CO_2$, pH 7.3), and allowed to recover in the same buffer at 37 °C for 60 min[65]. During imaging, slices were constantly perfused with 37 °C modified (epileptogenic) saline (37 °C) containing 125 NaCl, 5 KCl, 25 glucose, 1.25

**Fig. 10 Multi-plane imaging and optogenetics for functional circuit mapping. a, b**, DL (**a**) and nTC₁ (**b**) 1024 × 1024 px scans of the ventral nerve cord of a 3rd instar (L3) VGlut:GCamP6f *Drosophila* larva. **c–f** Scan-profiles taken in DL (**c**) and nTC₁ (**d**) across 6 planes spaced ~15 μm apart. **e, f** Schematic of first instar (L1) elav:GCamP6s; Ocro:CsChrimson *Drosophila* larva from top (**e**) and side (**f**), with CsChrimson (red) and GCaMP6s (green) expression pattern and scan-planes indicated. **g–k** optogenetic circuit mapping of olfactory processing centres across the larval brain. Six scan planes (170 × 340 px each) were taken at 0.98 Hz/plane (i.e. volume rate) during the presentation of 587 nm light flashes (2 s) to activate CsChrimson in olfactory sensory neurons (OSNs). Brain anatomy (**g**) and false-colour coded fluorescence difference image (**h** 1–2 s after flash onset minus 1–2 s prior to flash onset), with fluorescence activity traces (**i** raw and **j** event triggered average). For a zoom-in on the antennal lobe in a different specimen, see also Fig. S5. **k** data from (**h**) summarised: top right: max-projection through the brain, with left and bottom showing transverse max-projections across the same data-stack. VNC, ventral nerve cord. AL, antennal lobe, L(H)/M(B), Lateral horn/Mushroom body, CL, Contralateral (antennal) lobe.

---

NaH₂PO₄, 26 NaHCO₃, 2 CaCl₂. Brain slices were imaged at 930 nm and 100–150 mW.

**Mouse surgical procedures for in vivo-imaging of the barrel cortex.** Head bar implantation surgery has been described elsewhere[66]. Briefly, under aseptic conditions, a male mouse expressing a calcium indicator in pyramidal neurons (GCaMP6f; GP5.17[64]) was anaesthetised with isoflurane and implanted with a custom-made head bar. A circular 3 mm diameter craniotomy centred at 3.0 mm lateral and 1.0 mm posterior to bregma was made to expose the cranial surface. A cranial window, consisting of a 3 mm circular coverslip and a 5 mm circular coverslip (Harvard Apparatus), was placed over the craniotomy and secured in place with cyanoacrylate tissue sealant (Vetbond, 3 M). Following 7 days of recovery, the mouse was handled daily and acclimated to a head fixation apparatus over a treadmill for a further 9 days. During 2P imaging, the head-fixed mouse could locomote freely on a custom-made treadmill. The mouse was awake and received fluid rewards between imaging batches. Cortical neurons were imaged at 960 nm and 100–150 mW.

**Drosophila larval preparation and in-vivo imaging.** Flies were maintained at 25 °C in 12 h light:12 h dark conditions. Fly stocks were generated using standard procedures. The genotypes of the *D. melanogaster* flies used were: *elav-Gal4; LexAOp-CsChrimson* and *w; UAS-GCaMP6s; Orco-LexA*. These two strains were crossed to each other (collecting virgins from the first one and males from the second one) and placed on laying-pots at 25 °C for larval collection. The laying-pots had a grape juice agar plate with an added drop of yeast paste supplemented with all-trans retinal (Sigma-Aldrich) to a final concentration of 0.2 mM. Yeast supplemented agar plates were changed every day and first instar larvae were picked off the new changed plate. First instar larvae were collected from yeast supplemented agar plates and dissected on physiological saline as in ref. [67] (in mM): 135 NaCl, 5 KCl, 5 CaCl₂-2H₂O, 4 MgCl₂-6H₂O, 5 TES (2-[[1,3- dihydroxy-2-(hydroxymethyl)propan-2-yl]amino]ethanesulfonic acid), 36 Sucrose, adjusted to pH 7.15 with NaOH. Larvae were dissected to expose the brain while maintaining intact the anterior part of the animal and the connection between OSN cell bodies and the brain, subsequently one of the olfactory nerves was cut with the forceps. The preparation was then positioned on top of a freshly coated with poly-lysine (Sigma-Aldrich, P1524-100MG), and covered in 2% low melting point agarose (Fisher Scientific, Cat: BP1360-100) diluted in physiological saline, to prevent movement associated with mouth-hook contractions. The sample was then submerged in physiological saline. Larval brains were imaged at 930 nm and 30–60 mW.

**User manual.** A complete user manual for the nTC design, as well as a bill of materials (BOM), 3D printable lens holders and printed circuit board (PCB) designs are available online at https://github.com/BadenLab/nTCscope.

**DL 2P microscope.** Our setup was based on a Sutter MOM-type two-photon microscope (designed by W. Denk, MPI, Martinsried; purchased through Sutter Instruments) as described previously[68].

**Excitation path.** The excitation beam was generated by a tuneable femtosecond Ti:Sapphire laser (Coherent Vision-S, 75 fs, 80 MHz, >2.5 W). The laser passed an achromatic half-wave plate (AHWP05M-980, Thorlabs) and was subsequently equally split to supply two independent 2P setups using a beam-splitter for ultrashort pulses (10RQ00UB.4, Newport). Next, the beam passed a Pockels cell (350-80 with model 302 driver, Conoptics), a telescope (AC254-075-B and AC254-150-B, Thorlabs), and was finally reflected into the head part of Sutter MOM stage by a set of three silver mirrors (PF10-03-P01). We used a pair of single-axis galvanometric scan mirrors (6215H, Cambridge Technology) which directed the beam into a 50 mm focal length scan lens (VISIR 1534SPR136, Leica) at a distance of 56.6 mm. A 200 mm focal length tube lens (MXA22018, Nikon) was positioned 250 mm further along the optical path. From here, the now collimated excitation beam was directed onto the xyz-movable head of the Eyecup scope[68] which was controlled by a motorized micromanipulator (MP285-3Z, Sutter Instruments). Here, the beam was reflected by two silver parabolic mirrors to pass the collection

path dichroic mirror (T470/640rpc, Chroma) to finally slightly overfill the back aperture of the objective (Zeiss Objective W "Plan-Apochromat" ×20/1.0), thus creating a diffraction-limited excitation spot at the objective's nominal working distance of 1.8 mm. The distance between the tube lens and the objective's back aperture was 95 mm at the centre position of the xyz displacement mechanism, and the parabolic mirrors ensured that the optical excitation axes stayed aligned during movements of the microscope head.

**Collection path.** Collection was exclusively through the objective. For this, a dichroic mirror (T470/640rpc, Chroma) was positioned 18 mm above the objective's back aperture to reflect fluorescence light into the collection arm. Here, a 140 mm focal length collecting lens was followed by a 580-nm dichroic mirror (H 568 LPXR, superflat) to split the signal into two wavebands. The "green" and "red" channels each used a single-band bandpass filter (ET525/50 and ET 605/50, respectively, Chroma) and an aspheric condenser lens (G317703000, Linos) to focus light on a PMT detector chip (H10770PA-40, Hamamatsu).

**Image acquisition.** We used custom-written software (ScanM, by M. Mueller, MPI, Martinsried and T. Euler, CIN, Tuebingen) running under IGOR pro 6.3 for Windows (Wavemetrics) to control the setup. For hardware-software communication we use two multifunction I/O devices (PCIe-6363 and PCI-6110, National Instrument). Within ScanM, we defined custom scan-configurations: 1024 × 1024 and 512 × 512 pixel images with 2 ms per line were used for high-resolution morphology scans, while faster, 1 ms or 2 ms linespeed image sequences with 256 × 256 (3.91 Hz), 128 × 128 (7.81 Hz), 340 × 170 (5.88 Hz) or 128 × 64 (15.6 Hz) pixels were used for activity scans. All scans were unidirectional, and the laser was blanked via the Pockels cell during the turnarounds and retrace. This period was also used for light stimulation (zebrafish visual system and *Drosophila* optogenetics, see below).

**Non-collimated 2P microscope modifications.** We used two sets of modifications (nTC1 and nTC₂) to de-collimate the excitation path to different degrees. For nTC₁ (FOV 1.2–1.8 mm) we modified the original Sutter-MOM scan lens (VISIR 1534SPR136, Leica) by removing the second lens (i.e. the one closer to the tube lens) from the compound mount which changed the focal length from 50 to 190 mm. Alternatively, the entire de-constructed scan lens could also be replaced by a similar power off-the-shelf plano-convex lens. Our 190 mm lens (L1) was placed exactly 190 mm in front of the tube lens (so shifted 60 mm forward from its original position). Next, we introduced an additional plano-convex 175 mm focal distance lens (L2) (LA1229, Thorlabs). L2 was held in place by custom 3D printed mount (cf. user manual) inside the MOM's tube-lens holder and positioned anywhere between 0 and 100 mm in front of the tube lens. Depending on the exact position of L2 within this range, the effective FOV at the image plane could be adjusted between 1.2 mm (100 mm distance) to 1.8 mm (L2 and tube lens almost touching).

For nTC₂ (FOV 2.5–3.5 mm), we replaced the original scan lens with a single, 200 mm focal length plano-convex lens L3 (LA1708, Thorlabs). Like L2 in nTC₁, L3 was mounted on the same custom 3D printed holder and positioned anywhere within a distance of 0–100 mm in front of the tube lens. In this case the FOV at the image plane could be adjusted between 2.5 mm (100 mm distance) to 3.5 mm (L3 and tube lens almost touching). For detailed instructions including photos of the optical path, consult the user manual.

We selected lens types and positions based on the available space within the Sutter MOM head such that for nTC₁ and nTC₂, the IFP was always located in front of or behind the TL, respectively. However, depending on the design of a given 2P setup's excitation path, numerous alternative configurations are possible. Here, a straight-forward means to rapidly estimate the nature and scale of a given modification is to use a fluorescence test-slide and observe the change in working distance and FOV as the scan path is modified.

**Electrically tunable lens (ETL) for rapid axial focussing.** For rapid z-focussing we added a horizontal ETL (EL-16-40-TC-20D, Optotune) into the vertical beam path after the silver mirror that reflected the excitation beam up into the MOM head, 200 mm in front of the scan-mirrors. To drive the ETL we used a custom

current driver controlled by an Arduino Duo microcontroller (see user manual), capable of generating positive currents between 0 and 300 mA. The Arduino Duo received a copy of the scan-line command and in turn output commands to the current driver to effect line-synchronised changes in ETL curvature. Prior to initiating a scan, the specific to-be-executed Arduino programme was uploaded to the Arduino via serial from a PC running a custom Matlab-script (Mathworks). This Matlab script launched a simple graphical user interface (GUI) that allowed the user to configure the exact lens-path during a custom scan (see user manual). Accordingly, ETL control remained flexible and fully independent of the scan software. In this way, our solution can be readily integrated with any 2P system without need to change the software or acquisition/driver hardware. Notably, this ETL implementation can also be used by itself, without need for implementing any of the other optical adjustments described in this work. However, depending on the system's optics, the effective range of z-travel would likely be smaller. A detailed step-by-step guide to implement the ETL, including the control software and hardware is provided in the user manual.

**Pockels cell**. To control excitation laser intensity, we use a Pockels cell (Model 350-80, Conoptics; driver model 302, Conoptics). A line-synchronised blanking signal was sent from the DAQ to the drive to minimise laser power during the retrace. In addition, a custom circuit allowed controlling effective laser brightness during each scan line via a potentiometer (see user manual, designed by Ruediger Bernd, HIH, University of Tübingen). As required, this amplitude-modulated signal could then be further modulated by a second Arduino Due controlled by a standalone Matlab GUI to automatically vary effective laser power as a function of scanline index. In this way, laser power could be arbitrarily modulated on a line by line basis, for example, to compensate for possible power loss when imaging at increased depth.

**Light stimulation**. For visual stimulation of zebrafish larvae (Figs. 3, 5–7, Figs. S2, 4) we used a full-field, broadband spot of light projected directly onto the eyes of the fish from the front via a liquid light guide (77555, Newport) connected to a custom collimated LED bank (Roithner LaserTechnik) with emission peak wavelengths between 650 and 390 nm to yield an approximately equal power spectrum over the zebrafish's visual sensitivity range (described in detail in ref. [69]). LEDs were line-synchronised to the scanner retrace by an Arduino Due. For CsChrimson activation (Fig. 7) we used a custom 2 P line synchronised LED stimulator (https://github.com/BadenLab/Tetra-Chromatic-Stimulator) equipped with four 587 nm peak emission LEDs embedded in a custom 3D printed recording chamber.

**Image brightness measurements**. We imaged a uniform fluorescent sample consisting of two microscopy slides (S8902, Sigma-Aldrich) encapsulating a drop of low melting point agarose (Fisher Scientific, BP1360-100) mixed with low concentrated Acid Yellow 73 fluorescein solution (F6377 Sigma-Aldrich). Shown is the average brightness over the radius from the centre to the edge of the FOV (Matlab, custom scripts).

**PSF measurements**. We used $0.175 \pm 0.005\,\mu m$ yellow-green (505/515) fluorescent beads (P7220, Invitrogen) embedded in a 1 mm depth block of 1% low melting point agarose (Fisher Scientific, BP1360-100). Image stacks were acquired across $30 \times 30\,\mu m$ lateral field of view with $256 \times 256$ pixels resolution (0.12 μm/pixel) and 0.5 μm axial steps/frame. Laser power at the sample was 15–20 mW. For xy and z-dimensions, we calculated the full width at half maximum (FWHM) from Gaussian fits to the respective intensity profiles. Measurements were taken from set of the beads distributed across the entire FOV, and presented results are averages of at least 10 measurements of different beads, with error bars given in s.d.

To film the PSF and effective scan-plane(s) we focussed an air-objective (Plan Apo 4x/0.20, Nikon) onto the excitation spot elicited in a plastic cuvette with fluorescein (F2456 Sigma-Aldrich) dissolved in water which was positioned beneath the excitation objective. The camera path was fitted with a single-band bandpass filter (ET525/50, Chroma) and a colour CCD camera (Manta G-031C, Allied Vision). The camera was controlled with its dedicated software (VIMBA, Allied Vision). Note that this technique as implemented here tended to overestimate PSF sizes (e.g. due to the low N.A. objective, and because the cuvette was slightly vertically tilted relative to the camera axis to prevent back-reflections) and was therefore not suitable to determine absolute PSF dimensions – rather it was intended to enable relative comparisons of PSFs across scan configurations and positions. All quantitative PSF measurements were instead performed on beads (see above).

**Reporting summary**. Further information on research design is available in the Nature Research Reporting Summary linked to this article.

## Data availability
All systematic data generated in this study have been deposited on Zenodo (https://doi.org/10.5281/zenodo.5769794), and are further available in the Source data file. Source data are provided with this paper.

## Code availability
Code used in this work is available on Zenodo/Github https://doi.org/10.5281/zenodo.5763750.

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

## Acknowledgements

We thank Sabi Abdul-Raouf Issa for providing the VGlut:GCaMP6f *Drosophila* sample, and John Bear for helping with the generation of the Islet2b:mGCaMP6f line. The authors would also like to acknowledge support from the FENS-Kavli Network of Excellence and the EMBO YIP. Funding was provided by the European Research Council (ERC-StG "NeuroVisEco" 677687 to T.B., ERC-StG "EvolutioNeuroCircuit" 802531 to L.L.P.G.), The Wellcome Trust (Investigator Award in Science 220277/Z20/Z to T.B.), The UKRI (BBSRC, BB/R014817/1 to T.B., BB/S00310X/1 to K.S., and MRC, MC_PC_15071 to T.B. and M.M., MR/P006639/1 to M.M. and MR/P010121/1 to K.S.), the Leverhulme Trust (PLP-2017-005 to T.B.), the Lister Institute for Preventive Medicine (to T.B.), the Marie Curie Sklodowska Actions individual fellowship ("ColourFish" 748716 to T.Y.) from the European Union's Horizon 2020 research and innovation programme, and the German Research Foundation (DFG) through Collaborative Research Center CRC 1233 (project number 276693517, to T.E.). LLPG's research was supported by the The Francis Crick Institute.

## Author contributions

F.K.J. and T.B. designed the study, with inputs from T.E. and all authors; F.K.J. implemented and tested hardware and software modifications, with input from P.B., T.Y., T.B. and T.E. F.K.J. and T.B. analysed the data, with inputs from all authors. P.B. assisted with hardware and software testing and troubleshooting and built the visual stimulator. M.R.B. and M.M. provided mice for in vivo imaging and assisted with their handling and imaging. T.Y. and M.Z. generated Islet2b:mGCaMP line and assisted with zebrafish sample preparation and testing. E.K. and K.S. provided mouse brain acute slice samples and assisted with handling and imaging. L.L.P.G. provided *Drosophila* sample and assisted with handing and imaging. T.B. built the optogenetics stimulator. F.K.J. and T.B. wrote the manuscript with inputs from all authors.

## Competing interests

The authors declare no competing interests.

**Additional information**

