## [Peer Review File · Nature Communications]

Non-Telecentric two-photon microscopy for 3D random access mesoscale imagingEditorial Note: This manuscript has been previously reviewed at another journal that is not operating a transparent peer review scheme. This document only contains reviewer comments and rebuttal letters for versions considered at *Nature Communications*.

REVIEWER COMMENTS

Reviewer #1 (Remarks to the Author):

The revised manuscript by Janiak FK et al. entitled "Non-Telecentric 2P microscopy for 3D random access mesoscale imaging at single cell resolution" has addressed many questions I raised during the first round of review. The technical documentation is much more detailed now and the system performance measurement as for the PSF, FOV and field curvature are solid. Now I am convinced by the fact that this non-telecentric design indeed made the fast, long-distance xy- and z- ROI jumping much easier, which may be the most advantage in this work. Before I would finally suggest it to be published in nature communication, I still have a few questions:

1) A key, convincing evidence for single-cell-resolution imaging with nTC2, 3.5mm FOV.

The new off-the-shelf long-magnification, high-resolution objectives have made the ~1mm FOV single-cell-resolution imaging very easy now. For example, the 2P microscope in our lab equipped with the Thorlabs objective TL10X-2p can easily image 1,000 of neurons in 1mmx1mm FOV with sub-cellular-resolution. And it seems to be cheaper than the objective authors used in their work (Zeiss, 20X 1.0). But it cannot achieve 3.5 mm FOV as Janiak FK et al claimed here, which may be the most important advantage people will appreciate from their work. However, from the whole manuscript, I cannot find a clear evidence that the author indeed resolved single cell in 3.5mm FOV (nTC2). The reason may be that the low imaging speed limited the number of pixels authors used (512x512 for ~1 Hz) so that the pixel size (~70 um/pixel) was even bigger than single cell. The zoom-in images in Fig1 d-f were the indirect evidences but were not very informative because by zooming into small ROIs, people cannot tell if the resolution was the same across the whole FOV or just high in the center of the FOV.

So the author should either replace one Galvo mirror with a resonant scanner to increase the frame size without losing the temporal resolution (at least 2048x2048) and show they can resolve single-cell dynamic in the full 3.5mm FOV, or do a structure imaging using static indicator (like Thy1-GFP) with slower imaging speed (2048 with 0.25 Hz) so they can show more details of soma and dendrites in the full 3.5mm FOV.

2) I suggest the authors to add one more video related to the Fig2c,2d. So people can have a more direct comparison of the imaging quality in different configurations.

3) How deep can you go in mouse brain? By using the Thy1-GCaMP line, most of the neurons have soma in L5 and terminals in L1. Authors showed the dynamics of dendrites (video 8) which must be from the superficial L1. Can you also see the soma in L5? Otherwise, can you use other lines like CaMKII- or hSyn- to image L2/L3 neurons?

4) How to switch between DL, nTC1 and nTC2? Do you need to manually remove and re-mount lenses? Or is there an easier, software-controlled mechanism? This is important if you really claimed that "the end users can easily switch between different modes for different applications".

Reviewer #2 (Remarks to the Author):

This revised version of the manuscript is greatly improved and the authors did a good job in addressing most points raised in review. I particularly appreciated the clarification regarding the overall aim of the paper (i.e. to present a cheap and easily implementable way to flexibly control FOV and PSF dimension in two-photon imaging) and the discussion of system performance compared to other published two-photon mesoscopic imaging systems. The authors added important new data to characterize the optical properties of the system, including the evaluation of spatial resolution under various experimental conditions and across the FOV, and more convincing imaging of large FOVs. I have a few but important points that should be addressed to further improve the manuscript. Specifically,

1) The title, abstract, and the main text contain claims of single cell resolution. In their response to previous referees the authors stated that: "As noted by the reviewer, whether or not this can be achieved depends as much on the specific PSF dimensions as it does on the biological sample. If labelling is sufficiently sparse, single cells will be easily picked up even with a large PSF, while if labelling is dense (and structures are small) even a DL approach is rapidly limited in this regard." I agree with what the authors stated in their response. For the very same reason used by the

authors, I thus think that, in the context of the present study, claims of single cell resolution should be explicitly framed as follows. The system can achieve single cell resolution under specific experimental conditions which do not necessarily only depend on the microscope performance (e.g. sparse labeling of the sample). With the extended dimension of the axial PSF, the nTC system clearly cannot achieve single cell resolution under other specific conditions (e.g. less sparse labeling). I believe this is an important point to be addressed and I think it would be a pity to have this claim, which could sound like a strong overstatement for many people in the field, in an otherwise sound and complete study. I would suggest to remove single cell resolution from the title. Moreover, on lines 24, 120-121, 358-359, 527, 649, 773 I would suggest to explicitly state that the system can achieve single cell resolution only when the staining is sufficiently sparse to allow this.

2) Lines 207: it would be helpful to better explain what the authors mean by saying "simplified optical path". Similarly, on line 290 the authors state that the nTC approach "uses fewer optical elements in the laser path". Can you please explain what do you mean by saying "fewer"? In nTC1 configuration the scan-lens (1 optical element) is substituted with 2 lenses (L1 and L2). In the nTC2 configuration, the scan-lens is substituted with L3. A clarification seems needed.

3) Lines 309-311: "Accordingly, increasing the FOV using nTC mainly elongated the PSF, while restricting its lateral expansion to remain principally suitable for providing single cell resolution for the largest 3.5 mm expansion." This sentence seems to imply that single cell resolution can be achieved with good lateral resolution and an elongated PSF. However, in the discussion (line 844-846) the authors state: "Notwithstanding, any axial expansion in the PSF must be balanced with potentially undesirable merging of distinct image structures separated in depth". Based on the argument the authors provided in their response to the comments of the previous reviewers regarding the achievement of single cell resolution, i.e. to match the resolution of the scope with the properties of the sample, the sentence on page 309-311 should be modified.

Minor:

4) Line 409: "...could be collected..." .

Reviewer #4 (Remarks to the Author):

This manuscript reports an interesting exploration towards mesoscale 2PF imaging without the need for a customized and likely oversized objective lens. Instead, an off-the-shelf infinity-corrected objective is used in combination with a non-collimated, diverging input beam so as to extend the focal plane (which could be curved) further from the lens and enlarge the sample-side field of view. Substantial experimental data on zebrafish, mice and fruit flies are provided to demonstrate the applicability of this approach.

While this so-called non-telecentric configuration is an interesting and potentially valuable imaging approach, I have some major concerns, particularly with the claims made by the authors and their quantification of the imaging system's performance. I feel strongly that the authors need to be more objective about their approach and address the following comments with further substantial revisions and/or modifications in order for this manuscript to be acceptable for publication.

Major concerns --- Technology

While this non-conventional usage of an infinity-corrected objective has some merits, the concurrent shortcomings should also be acknowledged and carefully characterized to present an objective and non-overselling picture. Some conclusions claimed in the current manuscript also lack solid theoretical or experimental support, as detailed below:

1. FOV size: throughout the manuscript the FOV of the Zeiss W Plan-Apochromat 20X 1.0 NA objective is quoted to be ~0.5 mm in diameter in a diffraction-limited (DL) sense, which serves a base FOV against which new configurations are compared. However, since the expanded FOVs in the nTC setups are clearly non-diffraction limited, it seems only fair to compare them to the "usable" ~1-mm-diameter FOV of the 20X objective. Also it is not clear how the DL range is determined to be 0.5 mm here; in Supplementary Fig. S1c, the brightness curve for 0.5 mm DL

case seems to be truncated at 0.25 mm radius. I suggest the authors at least extend this curve to the radius where the brightness starts to plunge as the other three cases, and come up with a common criterion, say 50% normalized brightness, for FOV diameter.

2. PSF measurement with beads: the statement "PSF-dimensions are affected by a myriad of additional factors such as laser wavelength and power, as well as the specific measurement method (e.g. bead types)" (main text, line 318-320) does not comply with basic physics and optics. While PSF measurement, as any other experiments, needs to be carried out with sufficient rigor, the PSF size itself should not depend upon incident power or bead sizes, etc.

a. Especially, in Supplementary Fig. S1g-i, the PSF sizes were reported to depend on incident power (grows with power, to be more specific). While cranking up the incident power increases the excitable region in the beads phantom (i.e. position further away from the focus gets also sufficient power for 2P excitation), this expanded signal-generating zone is not a PSF. Instead, the PSF dimension is determined only by optics and should be reported with appropriate incident power in a normalized manner (e.g. FWHM). Please do re-check the beads data and re-do PSF measurements if needed.

b. Supplementary notes, line 5-8, paragraph 1, page 1: "the 2P cross-section, is for most fluorescent beads different from that of most popular biosensors for reporting neural activity". While this statement is true itself, it has little to do with why beads measurements do not tell fully the practical resolution in tissue. It is tissue scattering, not the difference in 2P cross section, that plays a fundamental role in the latter case. Please revise the argument accordingly.

c. In summary, for an incoherent imaging system, reporting the PSF using small enough beads is theoretically sound and gold standard in practice. It tells the best achievable resolution in the absence of scattering and can be compared across publications (against the authors' argument in response to review #2, on page 13).

d. The PSF values reported are not quite consistent. In Fig. 1n, the lateral PSF is reported to be ~ 1.0 μm for nTC1, and ~ 2.0 μm for nTC2. In Fig. 2b, one can easily estimate that the lateral PSF for the 3.5-mm-FOV case is much larger than 2.0 μm , when comparing to the 500 μm scale bar. Copying this image into ImageJ, and the lateral FWHM measures ~ 15 μm at least. This discrepancy should be addressed.

e. Related to this point is that reviewer #3 asked how different the PSF is off axis at peripheral locations, but the authors answered only about axial PSF. It is equally important to characterize the degradation in lateral PSF at the periphery of the FOV, especially considering that this unconventional use of a commercial objective breaks down the Abbe sine condition according to which commercial high-NA objectives are designed. As already requested by Reviewer #1, I would also urge the authors provide careful ZEMAX simulation and/or bead-based PSF characterization throughout the FOV for each nTC setup.

3. For PSF measurement from the side using a 4X 0.2 NA air-objective and a camera (Fig. 2a-b): the caveat is that the smallest that can be resolved by this approach is inherently limited by i) the theoretical lateral resolution of the 0.2 NA objective; ii) aberration induced by the complicated water-plastic cuvette wall-air interface (no info on how thick the water and plastic layers are); iii) sampling density determined by tube lens and camera pixel size (no details given here); and iv) the limited depth-sectioning capability (or axial resolution) of this essentially wide-field measurement. Therefore, the accuracy of this approach is definitely insufficient for the DL case, and the extent to which it affects those nTC cases mandates further consideration and careful characterization.

4. Signal enhancement: at least two types of signal boost were discussed in the manuscript, i.e. "nTC can yield an overall signal boost of up to ~ 18 -fold with an above-stage detector (or ~ 4 - 5 -fold at equal laser power on the sample)" (line 217-219, main text). The logic here is the under-filling associated with nTC setups brings a higher percentage of laser power to the sample (as reported in Fig. 1j) and thus "this signal loss was greatly outweighed by the increased overall availability of photons for collection in the first place (relative to DL: ~ 10 - 20 -fold for nTC1 and 16-18 for nTC2 and 2)" (line 424-426, main text). There are several questions that I suggest the authors to explain:

a. The improved power percentage to the sample, albeit important in practice for deep 2P imaging, should not be factored into the calculation of signal boost. It is more meaningful and fair to compare signal for given incident power onto the sample, since eventually it is photo- and thermal

damage in the sample, not in the objective lens, that limits the highest power to use.

b. More importantly, the concept that an enlarged excitation volume generates more signal is questionable. Based on fundamental principles of 2P excitation, when the effective focusing NA decreases by M -fold and thus the lateral focal area growing by M^2 -fold, the excitation beam intensity decreases by $1/M^2$, and thus the 2PE efficiency decreases by $1/M^4$. When focusing into an infinite thick uniform sample (e.g. fluorescein solution cuvette), this $1/M^4$ deficiency in excitation efficiency can be compensated by the correspondingly expanded focal volume, i.e. M^2 expansion in lateral area and M^2 elongation of axial range (see Chris Xu [1]). But for real-world biological tissues, especially sparsely labeled neurons, the biosensor distribution is clearly not uniform, and therefore the significantly expanded axial PSF (up to 40 μm) in the nTC setup might NOT encounter biosensor and generate 2P signal across the entire axial range, and therefore yield less 2P fluorescence. Therefore, how the nTC setups, despite its collection loss, yield overall signal boost "even if keeping effective laser power on the sample constant" (line 431-432) is unclear and unconvincing. I urge the authors conduct rigorous experiments (e.g., imaging into the same fluorescein solution with identical pixel dwelling time, and compare the obtainable signal per pixel area) to quantitatively prove the extent of signal boost. Otherwise this claim should be removed.

c. One caveat is that the authors are actually describing the final image brightness, where the scan step (i.e. digital pixel) is much finer than the lateral PSF so that the collected photons (i.e. the reported brightness) at each digital pixel is essentially photons from all neighboring digital pixels binned together. I would like to emphasize that: i) such enhancement in pixel brightness is not equal to signal boost, as the latter typically referred to 2P fluorescence intensity, i.e. photons per cross-sectional area per sec; ii) it is the emission intensity, not the emission power per se, that matters in practice; otherwise, one can also digitally bin a diffraction-limited image to boost the brightness in the cost of a blurred image. Please double-check and make the concept clear in the manuscript.

5. Aberration: as pointed out by Reviewer #1 already, no "new types" of aberration does not mean "no more aberrations". There are always 5 major types of monochromatic aberrations, and it is the amount of aberration that matters. So I suggest the authors re-phrase their argument about aberration type on line 817-818 in the main text. And,

a. In line 852, "reduced spherical aberrations" was claimed in the Conclusion part which is, however, against the common sense that spherical aberration grows with defocusing. It is possible that in such nTC setups, the underfilling of back aperture reduces the effective entrance pupil size and subsequently the overall spherical aberration, but detailed quantitative evidence is needed to support this claim.

b. Besides field curvature, another expectable important aberration is distortion in FOV, as also mentioned by Reviewer #2. Note that we are not talking about distorted PSF, but distortion aberration regarding how much the lateral magnification changes for a given focal plane. Please reconsider the question and provide clear, rigorous characterization.

6. Finally, the term "non-telecentric" is not a very appropriate here either. The reason is that telecentricity refers to collimated chief ray and requires relaying the galvo (using scan and tube lens) to exactly the back focal plane of an objective. In standard 2P microscopes, however, the galvo is often relayed to the back aperture, rather than the back focal plane, for maximal power throughput. Therefore, common 2P microscopes are already non-telecentric. So I suggest the authors rethink about this and probably choose a better term.

Major concerns --- Applications

7. The top concern is that the PSF can grow up to 40 μm in the axial direction, which is not truly "single-cell resolution", as evidenced by the final images in Figs. 4-9 and pointed out by previous reviewers. The ability to see some individual isolated cells in sparsely labeled specimen is not equivalent to single-cell resolution. Again, the approach reported in this manuscript has its unique merits for cost-effective, mesoscale 2P imaging, but I suggest the author avoid overselling it as a "single-cell resolution" imaging solution.

8. Another big concern is a lot of images and videos shown here are too saturated (especially, Fig. 5d, 5k, 7d and associated supplementary videos). I suggest the authors try square root scaling or logarithmic scaling in image and video rendering to better handle the big dynamic range and bring more details up.

Minor suggestions

9. In line 521, "Fig. 2f" should be "Fig. 4f"?

10. In Fig. 6d, the two images corresponding to 200-um and 250-um peak displacement, the central region of the curved plane, illustrated by a red curved line, is already outside the cartoon fish head, but the corresponding 2P images are not void in the center. Please double check.

11. In Figs. 8j-k, it is unclear which axes correspond to x-, y-, or z-directions. And why the FOV is labeled to be 3.5 mm in Fig. 8k, but the image is 200 um in dimensions. Please further label and explain in the caption and main text.

12. Line 850, PSF tailoring is not truly demonstrated but only proposed in this manuscript, so it should not be claimed in the Conclusion session. Instead, the curved scan profile as reported in Fig. 6 could be mentioned here.

References

1. Xu, C. and W.W. Webb, Multiphoton Excitation of Molecular Fluorophores and Nonlinear Laser Microscopy, in *Topics in Fluorescence Spectroscopy: Volume 5: Nonlinear and Two-Photon-Induced Fluorescence*, J.R. Lakowicz, Editor. 2002, Springer US: Boston, MA. p. 471-540.

Reviewer #1

The revised manuscript by Janiak FK et al. entitled “Non-Telecentric 2P microscopy for 3D random access mesoscale imaging at single cell resolution” has addressed many questions I raised during the first round of review. The technical documentation is much more detailed now and the system performance measurement as for the PSF, FOV and field curvature are solid. Now I am convinced by the fact that this non-telecentric design indeed made the fast, long-distance xy- and z- ROI jumping much easier, which may be the most advantage in this work.

Thank you.

Before I would finally suggest it to be published in nature communication, I still have a few questions:

1) A key, convincing evidence for single-cell-resolution imaging with nTC2, 3.5mm FOV.

The new off-the-shelf long-magnification, high-resolution objectives have made the ~1mm FOV single-cell-resolution imaging very easy now. For example, the 2P microscope in our lab equipped with the Thorlabs objective TL10X-2p can easily image 1,000 of neurons in 1mmx1mm FOV with sub-cellular-resolution. And it seems to be cheaper than the objective authors used in their work (Zeiss, 20X 1.0). But it cannot achieve 3.5 mm FOV as Janiak FK et al claimed here, which may be the most important advantage people will appreciate from their work. However, from the whole manuscript, I cannot find a clear evidence that the author indeed resolved single cell in 3.5mm FOV (nTC2). The reason may be that the low imaging speed limited the number of pixels authors used (512x512 for ~1 Hz) so that the pixel size (~70 um/pixel) was even bigger than single cell. The zoom-in images in Fig1 d-f were the indirect evidences but were not very informative because by zooming into small ROIs, people cannot tell if the resolution was the same across the whole FOV or just high in the center of the FOV.

So the author should either replace one Galvo mirror with a resonant scanner to increase the frame size without losing the temporal resolution (at least 2048x2048) and show they can resolve single-cell dynamic in the full 3.5mm FOV, or do a structure imaging using static indicator (like Thy1-GFP) with slower imaging speed (2048 with 0.25 Hz) so they can show more details of soma and dendrites in the full 3.5mm FOV.

For clarity, we have now removed the reference to single cell resolution from the title. In the abstract and throughout the text we have either removed this note, or where we have kept it, made clear that this depends on the specific sample.

Regarding the suggested experiments, COVID measures are limiting our access to mice to more mice due to, so it remains difficult to obtain a sample where only a handful of single neurons are labelled (e.g. following viral injection or similar). However, we do think that the many zebrafish samples, including those presented for the previous revision, as well as the many other examples and quantifications (now e.g. now many more PSF measurements in view of

comments raised by reviewer 4) serve to support all essential points raised in the MS.

2) I suggest the authors to add one more video related to the Fig2c,2d. So people can have a more direct comparison of the imaging quality in different configurations.

Thank you for this comment; we now provide the requested video as SVideo 3.

3) How deep can you go in mouse brain? By using the Thy1-GCaMP line, most of the neurons have soma in L5 and terminals in L1. Authors showed the dynamics of dendrites (video 8) which must be from the superficial L1. Can you also see the soma in L5? Otherwise, can you use other lines like CaMKII- or hSyn- to image L2/L3 neurons?

Unfortunately, we still do not have access to mice for in vivo experiments due to COVID measures. From the existing data however, we think it is quite clear that we can “usefully image” at some ~500 microns depth in cortex (see xyz scan in Fig. 9j/k, previously Fig. 8j/k), which is approximately in line with what might be expected from previous work. It is possible we can go still deeper, but this was not one of the goals of the paper, and we do not (intentionally) make any claims in this direction. Rather, we focus on the fact that our ETL allows us to jump sufficiently far to enable near-simultaneous imaging over this range. From the MS:

“Finally, we also recruited the ETL to set up an axially tilted scan plane. This allowed quasi-simultaneous recording from neurons separated several hundreds of μm in depth across layers 1-4 of the mouse cortex in vivo (Fig. 9j-l).”

Notably, we actually split our Vision-S laser for two setups, so in existing experiments the main limitation at depth was total available power (rather than e.g. photodamage). Conceivably we could of course direct the full power to one setup and redo the experiment, but then we are again faced with our continued lack of access to suitable samples. As these experiments would probably delay the paper by many more months, we rather hope that since imaging at depth is not a focus of our study, this could perhaps be considered as part of future work.

4) How to switch between DL, nTC1 and nTC2? Do you need to manually remove and re-mount lenses? Or is there an easier, software-controlled mechanism? This is important if you really claimed that “the end users can easily switch between different modes for different applications”.

Currently this is done manually, but literally involves switching/shifting one or two lenses. In our hands, this takes less than a minute, as demonstrated in Video S2. Of course, a user may choose to design a slider system to automate this process. We now clarify in the legend to Video S2 that: “Lens switching was done by hand”.

Reviewer #2

This revised version of the manuscript is greatly improved and the authors did a good job in addressing most points raised in review. I particularly appreciated the clarification regarding the overall aim of the paper (i.e. to present a cheap and easily implementable way to flexibly control FOV and PSF dimension in two-photon imaging) and the discussion of system performance compared to other published two-photon mesoscopic imaging systems. The authors added important new data to characterize the optical properties of the system, including the evaluation of spatial resolution under various experimental conditions and across the FOV, and more convincing imaging of large FOVs.

Thank you!

I have a few but important points that should be addressed to further improve the manuscript. Specifically,

1) The title, abstract, and the main text contain claims of single cell resolution. In their response to previous referees the authors stated that: "As noted by the reviewer, whether or not this can be achieved depends as much on the specific PSF dimensions as it does on the biological sample. If labelling is sufficiently sparse, single cells will be easily picked up even with a large PSF, while if labelling is dense (and structures are small) even a DL approach is rapidly limited in this regard." I agree with what the authors stated in their response. For the very same reason used by the authors, I thus think that, in the context of the present study, claims of single cell resolution should be explicitly framed as follows. The system can achieve single cell resolution under specific experimental conditions which do not necessarily only depend on the microscope performance (e.g. sparse labeling of the sample). With the extended dimension of the axial PSF, the nTC system clearly cannot achieve single cell resolution under other specific conditions (e.g. less sparse labeling). I believe this is an important point to be addressed and I think it would be a pity to have this claim, which could sound like a strong overstatement for many people in the field, in an otherwise sound and complete study. I would suggest to remove single cell resolution from the title. Moreover, on lines 24, 120-121, 358-359, 527, 649, 773 I would suggest to explicitly state that the system can achieve single cell resolution only when the staining is sufficiently sparse to allow this.

For clarity, we have now removed the reference to single cell resolution from the title. In the abstract and throughout the text we have either removed this note, or where we have kept it, made clear that this depends on the specific sample.

2) Lines 207: it would be helpful to better explain what the authors mean by saying "simplified optical path".

To address this point, we have now rephrased as: "The simplified optical path (i.e. fewer lenses between the scan mirrors and sample) and under-filling of the objective's back aperture means more laser power is available at the sample plane".

Similarly, on line 290 the authors state that the nTC approach “uses fewer optical elements in the laser path”. Can you please explain what do you mean by saying “fewer”? In nTC1 configuration the scan-lens (1 optical element) is substituted with 2 lenses (L1 and L2). In the nTC2 configuration, the scan-lens is substituted with L3. A clarification seems needed.

The native scan lens of the Sutter system consists of two achromatic doublets, i.e. technically 4 optical elements, not one. In nTC, we replace it with one doublet and one singlet (3 elements, nTC1), or one singlet (1 element, nTC2).

(We appreciate that doublets can be considered single lenses, but technically they are two lenses that are cemented together. Moreover, even if approximated as a single lens, they are still rather thick and attenuate light more strongly than a singlet).

We added: “...laser path (by replacing the native scan lens consisting of two doublets with one doublet and one singlet in nTC₁, or one singlet in nTC₂)...”

3) Lines 309-311: “Accordingly, increasing the FOV using nTC mainly elongated the PSF, while restricting its lateral expansion to remain principally suitable for providing single cell resolution for the largest 3.5 mm expansion.” This sentence seems to imply that single cell resolution can be achieved with good lateral resolution and an elongated PSF. However, in the discussion (line 844-846) the authors state: “Notwithstanding, any axial expansion in the PSF must be balanced with potentially undesirable merging of distinct image structures separated in depth”. Based on the argument the authors provided in their response to the comments of the previous reviewers regarding the achievement of single cell resolution, i.e. to match the resolution of the scope with the properties of the sample, the sentence on page 309-311 should be modified.

The statement on single cell resolution (“to remain principally suitable for providing single cell resolution even for the largest 3.5 mm expansion”) was deleted.

Minor:

4) Line 409: “...could be collected...” .

Fixed.

Reviewer #4

This manuscript reports an interesting exploration towards mesoscale 2PF imaging without the need for a customized and likely oversized objective lens. Instead, an off-the-shelf infinity-corrected objective is used in combination with a non-collimated, diverging input beam so as to extend the focal plane (which could be curved) further from the lens and enlarge the sample-side field of view. Substantial experimental data on zebrafish, mice and fruit flies are provided to demonstrate the applicability of this approach.

While this so-called non-telecentric configuration is an interesting and potentially valuable imaging approach, I have some major concerns, particularly with the claims made by the authors and their quantification of the imaging system's performance. I feel strongly that the authors need to be more objective about their approach and address the following comments with further substantial revisions and/or modifications in order for this manuscript to be acceptable for publication.

Major concerns --- Technology

While this non-conventional usage of an infinity-corrected objective has some merits, the concurrent shortcomings should also be acknowledged and carefully characterized to present an objective and non-overselling picture. Some conclusions claimed in the current manuscript also lack solid theoretical or experimental support, as detailed below:

1. FOV size: throughout the manuscript the FOV of the Zeiss W Plan-Apochromat 20X 1.0 NA objective is quoted to be ~ 0.5 mm in diameter in a diffraction-limited (DL) sense, which serves a base FOV against which new configurations are compared. However, since the expanded FOVs in the nTC setups are clearly non-diffraction limited, it seems only fair to compare them to the "usable" ~ 1 -mm-diameter FOV of the 20X objective. Also it is not clear how the DL range is determined to be 0.5 mm here; in Supplementary Fig. S1c, the brightness curve for 0.5 mm DL case seems to be truncated at 0.25 mm radius. I suggest the authors at least extend this curve to the radius where the brightness starts to plunge as the other three cases, and come up with a common criterion, say 50% normalized brightness, for FOV diameter.

We agree that it is possible to get a larger than 0.5 mm FOV with this objective (in fact that is what we show); however in this case, as noted by the reviewer, the system cannot be diffraction limited.

The 0.5 mm squared FOV is the maximal FOV of the chosen objective on a Sutter MOM DL configuration with $1/e^2$ overfilling of the back aperture (Helmchen and Denk 2005).

The plot noted by the reviewer (S1c) is "truncated" at 0.25 mm radius, because maximum safe mirror displacement has been reached. This is the factory-defined "intended" optical configuration of the Sutter-MOM. Nevertheless, the diagonal of the resultant square FOV is very close to being clipped by the objective back aperture (by factory design), meaning that the strictly diffraction limited FOV of this optical setup really is ~ 0.5 mm (or 0.6 mm at most). Demonstrating this point more directly would require overriding the maximal

safe displacement condition in our mirror commands, which we prefer to avoid because of the risk of damaging them.

2. PSF measurement with beads: the statement “PSF-dimensions are affected by a myriad of additional factors such as laser wavelength and power, as well as the specific measurement method (e.g. bead types)” (main text, line 318-320) does not comply with basic physics and optics. While PSF measurement, as any other experiments, needs to be carried out with sufficient rigor, the PSF size itself should not depend upon incident power or bead sizes, etc.

Apologies; this was a simple phrasing error. We intended to state that the estimation of PSF dimensions is dependent on all these factors (rather than the actual PSF). We have now clarified that in the MS.

For example, regarding wavelength, the PSF-size (for FWHM) in DL configuration is given in xy and z by (from Zipfel et al. 2003 Nat Biotech):

$$\mathbf{c} \quad \omega_{xy} = \begin{cases} \frac{0.320 \lambda}{\sqrt{2} \text{NA}} & \text{NA} \leq 0.7 \\ \frac{0.325 \lambda}{\sqrt{2} \text{NA}^{0.91}} & \text{NA} > 0.7 \end{cases} \quad \omega_z = \frac{0.532 \lambda}{\sqrt{2}} \left[\frac{1}{n - \sqrt{n^2 - \text{NA}^2}} \right]$$

Where λ is wavelength of the excitation laser, and NA is numerical aperture of the objective, and n is the refractive index of the sample.

Regarding intensity and bead types, we agree that the direct power dependence is small (<1%), however in practical terms the picture is more complex. A “true” PSF size can only be estimated if fluorescence intensity saturation is avoided, and this means that power must be kept minimal – however this minimal power level is routinely exceeded during imaging of neuronal structures. Of course, the degree of this problem strongly depends on the 2P cross-section of the sample (higher cross-section → increased inflation of effective PSF size as measured with FWHM). Beads have an unknown cross-section, but it is quite likely to be higher than e.g. GCaMP, accordingly, bead measurements have a tendency to overestimate the true PSF compared to typically used biosensors unless power is minimal to avoid the saturation effect. On the other hand, measuring a PSF using the biosensor itself (which would be the fairest approach) is not possible as we cannot generate a “GCaMP-point source” with current technology. Alternatively, one could measure a relevant biological sample with different power levels and look at z-sectioning. This would give a possibly useful impression of the degree of power-dependence in that specific application, but it would be very difficult to quantify, for example due to the inevitable superposition of excitation planes and also inhomogeneities in the sample.

Returning to the key points of our study – the demonstration that the FWHM of PSFs as measured with beads grows is a technical sidenote, rather than being specifically linked to the optical performance of our system. It is only intended as a demonstration of this often-overlooked problem. It means that directly comparing PSFs across studies (which we have been asked to do at some depth) is difficult unless all the above parameters (wavelength, intensity, bead type, embedding medium etc.) are known.

With the exception of the demonstration of the power dependence, where we went up to >100 mW (as routinely reached when imaging mouse cortex – although of course the CS of biosensors is lower than of beads), we measured all beads at 15-20 mW which is as close to minimal as can be achieved while still getting a usable signal. Accordingly, we think that the PSFs are as representative of the system's "true" performance as can be established.

a. Especially, in Supplementary Fig. S1g-i, the PSF sizes were reported to depend on incident power (grows with power, to be more specific). While cranking up the incident power increases the excitable region in the beads phantom (i.e. position further away from the focus gets also sufficient power for 2P excitation), this expanded signal-generating zone is not a PSF. Instead, the PSF dimension is determined only by optics and should be reported with appropriate incident power in a normalized manner (e.g. FWHM). Please do re-check the beads data and re-do PSF measurements if needed.

Please see our response to the above point.

b. Supplementary notes, line 5-8, paragraph 1, page 1: "the 2P cross-section, is for most fluorescent beads different from that of most popular biosensors for reporting neural activity". While this statement is true itself, it has little to do with why beads measurements do not tell fully the practical resolution in tissue. It is tissue scattering, not the difference in 2P cross section, that plays a fundamental role in the latter case. Please revise the argument accordingly.

We agree that scattering is an important aspect. However, the 2P cross section also matters, and it is much more directly related to (mis-)estimates from (usually agarose embedded) beads, which is the subject of this paragraph.

As detailed above, in our statement we intended to state that any differences in the 2P cross section between the beads and e.g. GCaMP can potentially lead to a misleading estimate of the excitation PSF.

To further clarify this, we have now added the following note at the end of the paragraph in question:

"Notwithstanding, the PSF also depends on the specific scattering in the sample (see "optical aberrations" below)."

We think that this will usefully guide the reader to the optical aberrations sections, where scattering is discussed at some depth.

c. In summary, for an incoherent imaging system, reporting the PSF using small enough beads is theoretically sound and gold standard in practice. It tells the best achievable resolution in the absence of scattering and can be compared across publications (against the authors' argument in response to review #2, on page 13).

We agree that the PSF measurement using beads is the best available gold standard and we did not mean to imply otherwise. We are in fact using it as such throughout the manuscript.

We are however simultaneously pointing out that we think that the excitation wavelength and power, as well as the 2P cross section of the sample also matter, alongside scattering, and that these can make cross-study comparisons of PSF sizes difficult. For this to be truly possible, it is important to use the same beads and medium, as well as matched effective laser power and wavelength, as discussed in the preceding comments. This remains the case even if power is kept low to ameliorate fluorescence saturation.

d. The PSF values reported are not quite consistent. In Fig. 1n, the lateral PSF is reported to be ~1.0 μm for nTC1, and ~2.0 μm for nTC2. In Fig. 2b, one can easily estimate that the lateral PSF for the 3.5-mm-FOV case is much larger than 2.0 μm , when comparing to the 500 μm scale bar. Copying this image into ImageJ, and the lateral FWHM measures ~15 μm at least. This discrepancy should be addressed.

The scan profiles estimated by the camera approach are not intended to provide numerical information on the real PSF sizes, as this will almost certainly be a gross overestimate. Rather, the goal of the camera approach was to establish a simple way to visualise how PSFs vary over the full field of view (e.g. field curvature, tilting etc, and how scans distribute across space in the varying configurations shown throughout the MS). This is now explicitly noted in the text, and we thank the reviewer for drawing our attention to this source of possible confusion:

“However, the resultant scan profiles overestimated PSF sizes and were thus not suitable to determine their absolute dimensions – rather, the goal was to observe relative variations in PSF shapes, positions and orientations over the full FOV.”

e. Related to this point is that reviewer #3 asked how different the PSF is off axis at peripheral locations, but the authors answered only about axial PSF. It is equally important to characterize the degradation in lateral PSF at the periphery of the FOV, especially considering that this unconventional use of a commercial objective breaks down the Abbe sine condition according to which commercial high-NA objectives are designed. As already requested by Reviewer #1, I would also urge the authors provide careful ZEMAX simulation and/or bead-based PSF characterization throughout the FOV for each nTC setup.

We now provide new experimental data to directly quantify PSF variations over the full field of view for all configurations (new Figure 2c-e). This directly shows that while the PSFs do indeed tilt a little at the edge (as already shown by the camera approach), they are not strongly distorted relative to the centre in any configuration. We hope that the new data noted above fully addressed this concern.

3. For PSF measurement from the side using a 4X 0.2 NA air-objective and a camera (Fig. 2a-b): the caveat is that the smallest that can be resolved by this approach is inherently limited by i) the theoretical lateral resolution of the 0.2 NA objective; ii) aberration induced by the complicated water-plastic cuvette wall-air interface (no info on how thick the water and plastic layers are); iii) sampling density determined by tube lens and camera pixel size (no details given here); and iv) the limited depth-sectioning capability (or axial resolution) of this essentially wide-field measurement. Therefore, the accuracy of this approach is definitely insufficient for the DL case, and the extent to which it affects those nTC cases mandates further consideration and careful characterization.

We agree that the camera generated scan profiles are not suitable to provide a quantitative measurement of absolute PSF dimensions. This is done by the beads (including now also the new added data beneath). We now discuss this further in the associated text:

“However, the resultant scan profiles overestimated PSF sizes and were thus not suitable to determine their absolute dimensions – rather, the goal was to observe relative variations in PSF shapes, positions, and orientations over the full FOV”

And again in the Methods section on PSFs:

“Note that this technique as implemented here tended to overestimate PSF sizes (e.g. due to the low N.A. objective, and because the cuvette was slightly vertically tilted relative to the camera axis to prevent back-reflections) and was therefore not suitable to determine absolute PSF dimensions – rather it was intended to enable relative comparisons of PSFs across scan configurations and positions. All quantitative PSF measurements were instead performed on beads (see above).”

4. Signal enhancement: at least two types of signal boost were discussed in the manuscript, i.e. “nTC can yield an overall signal boost of up to ~18-fold with an above-stage detector (or ~4-5-fold at equal laser power on the sample)” (line 217-219, main text). The logic here is the under-filling associated with nTC setups brings a higher percentage of laser power to the sample (as reported in Fig. 1j) and thus “this signal loss was greatly outweighed by the increased overall availability of photons for collection in the first place (relative to DL: ~10-20-fold for nTC1 and 16-18 for nTC2 and 2)” (line 424-426, main text). There are several questions that I suggest the authors to explain:

a. The improved power percentage to the sample, albeit important in practice for deep 2P imaging, should not be factored into the calculation of signal boost. It is more meaningful and fair to compare signal for given incident power onto the sample, since eventually it is photo- and thermal damage in the sample, not in the objective lens, that limits the highest power to use.

We agree that total power does not necessarily directly translate into a better signal. Accordingly, we also agree that this ~4-fold power gain is not necessarily part of the signal boost per se. This is why we explicitly note that signal boost is 4-5-fold if laser power is kept constant (rather than up to 18-fold) in our initial bullet point list. Moreover, when discussing reduced power loss, we note that this is potentially helpful if the goal is to image deep, or if multiple setups share a single laser.

Of course, as the PSF grows e.g. by using different nTC configurations, the power density experienced by a given image structure also changes, which makes it possible to increase laser power further before the sample “starts to burn”. Accordingly, the relationship between the best signal/noise that can be achieved and laser power / PSF shape is complex. We further elaborate on this point below.

b. More importantly, the concept that an enlarged excitation volume generates more signal is questionable. Based on fundamental principles of 2P excitation, when the effective focusing NA decreases by M-fold and thus the lateral focal area growing by M^2 -fold, the excitation beam intensity decreases by $1/M^2$, and thus the 2PE efficiency decreases by $1/M^4$. When focusing into an infinite thick uniform sample (e.g. fluorescein solution cuvette), this $1/M^4$ deficiency in excitation efficiency can be compensated by the correspondingly expanded focal volume, i.e. M^2 expansion in lateral area and M^2 elongation of axial range (see Chris Xu [1]). But for real-world biological tissues, especially sparsely labeled neurons, the biosensor distribution is clearly not uniform, and therefore the significantly expanded axial PSF (up to 40 μm) in the nTC setup might NOT encounter biosensor and generate 2P signal across the entire axial range, and therefore yield less 2P fluorescence. Therefore, how the nTC setups, despite its collection loss, yield overall signal boost “even if keeping effective laser power on the sample constant” (line 431-432) is unclear and unconvincing. I urge the authors conduct rigorous experiments (e.g., imaging into the same fluorescein solution with identical pixel dwelling time, and compare the obtainable signal per pixel area) to quantitatively prove the extent of signal boost. Otherwise this claim should be removed.

We agree that how a given PSF permits us to capture a given image structure (e.g. a cell body) depends on both the structure and the PSF, such that e.g. part of a large PSF will miss the structure and therefore the signal will stop increasing. Conversely, however, a large PSF is also much more likely to partly or fully capture a structure in the sample at all. Clearly this is a non-trivial relationship, and the real signal boost (or loss) will depend on the application.

This is part of the reason why we provide so many different example tissues and applications for the different configurations, alongside the direct quantification based on fluorescein-soaked tissue shown in the supplement (which seems very similar to the experiment suggested by the reviewer) as well as the direct comparison of the same set of neurons in the case of the zebrafish spinal cord. The latter, for example, shows that the likely most experimentally useful configuration for this particular sample is around nTC1 (1.8 mm) rather than the largest possible nTC2 configuration.

Nevertheless, we addressed the reviewer's concern and now also performed the suggested experiment. Specifically, we imaged the central point of each FOV in a cuvette filled with low concentration fluorescein solution (to not saturate the PMTs). The laser was kept constant at the level of the scan mirrors (hence more laser on the sample, as discussed), and the PMT gain and dwell time were also kept constant. We then compared the brightness readings from each of the two detectors for each nTC configuration against the readings from the DL configuration. This further confirmed that all nTC configurations provided a substantial overall signal boost on both detectors. This is now added to Figure S3d.

c. One caveat is that the authors are actually describing the final image brightness, where the scan step (i.e. digital pixel) is much finer than the lateral PSF so that the collected photons (i.e. the reported brightness) at each digital pixel is essentially photons from all neighboring digital pixels binned together. I would like to emphasize that: i) such enhancement in pixel brightness is not equal to signal boost, as the latter typically referred to 2P fluorescence intensity, i.e. photons per cross-sectional area per sec; ii) it is the emission intensity, not the emission power per se, that matters in practice; otherwise, one can also digitally bin a diffraction-limited image to boost the brightness in the cost of a blurred image. Please double-check and make the concept clear in the manuscript.

Please see previous response.

5. Aberration: as pointed out by Reviewer #1 already, no “new types” of aberration does not mean “no more aberrations”. There are always 5 major types of monochromatic aberrations, and it is the amount of aberration that matters. So I suggest the authors re-phrase their argument about aberration type on line 817-818 in the main text. And,

We agree, however the comment on “no new aberrations” stems from a previous round of reviews, and we feel it is important to explicitly note that new types of aberrations do not arise. Moreover, beyond the PSF expansion, which we think is extensively documented and discussed, we do not see any notable worsening in other types of aberrations (e.g. also see the new data on PSFs across the full FOV noted above, point 2e). Accordingly, we have rephrased as follows:

“In general, beyond the PSF expansion that results from bypassing the objective's infinity correction (Figs. 1,2, Fig. S1), the change from a standard 2P DL-setup to an nTC configuration does not bring about new types of

aberrations, nor does it notably worsen the inevitably pre-existing aberrations from the DL configurations”.

a. In line 852, “reduced spherical aberrations” was claimed in the Conclusion part which is, however, against the common sense that spherical aberration grows with defocusing. It is possible that in such nTC setups, the underfilling of back aperture reduces the effective entrance pupil size and subsequently the overall spherical aberration, but detailed quantitative evidence is needed to support this claim.

We followed the reviewer’s suggestion and have removed this statement on spherical aberrations.

b. Besides field curvature, another expectable important aberration is distortion in FOV, as also mentioned by Reviewer #2. Note that we are not talking about distorted PSF, but distortion aberration regarding how much the lateral magnification changes for a given focal plane. Please reconsider the question and provide clear, rigorous characterization.

We have now measured this on a fixed pollen sample by taking identical scans with different stage xy translations, followed by direct superposition of a corresponding image region. No evidence of distortion was observed. The data is added as Figure S2:

6. Finally, the term “non-telecentric” is not a very appropriate here either. The reason is that telecentricity refers to collimated chief ray and requires relaying the galvo (using scan and tube lens) to exactly the back focal plane of an objective. In standard 2P microscopes, however, the galvo is often relayed to the back aperture, rather than the back focal plane, for maximal power throughput. Therefore, common 2P microscopes are already non-telecentric. So I suggest the authors rethink about this and probably choose a better term.

Thank you for giving us the opportunity to clarify this point. We agree that some 2P setups are designed to use a mildly non-telecentric beam for the

aforementioned reasons. However, this usually comes about with a rather small angle of divergence where the goal, as noted, is to increase effective power available for imaging. Importantly, this is, by definition, no longer a diffraction limited system, as this requires the beam to be parallel (for an infinity corrected objective).

Here, we think that we have gone substantially beyond such systems, for example by explicitly varying the degree of divergence to much larger angles and quantified the consequences in depth. As such, we think that the term non-Telecentric usefully summarises our efforts, and we have thus opted to keep it.

Major concerns --- Applications

7. The top concern is that the PSF can grow up to 40 μm in the axial direction, which is not truly “single-cell resolution”, as evidenced by the final images in Figs. 4-9 and pointed out by previous reviewers. The ability to see some individual isolated cells in sparsely labeled specimen is not equivalent to single-cell resolution. Again, the approach reported in this manuscript has its unique merits for cost-effective, mesoscale 2P imaging, but I suggest the author avoid overselling it as a “single-cell resolution” imaging solution.

Following the reviewer’s advice, we have now removed the reference to single cell resolution from the title. In the abstract and throughout the text we have either removed this note, or where we have kept it, made clear that this depends on the specific sample.

8. Another big concern is a lot of images and videos shown here are too saturated (especially, Fig. 5d, 5k, 7d and associated supplementary videos). I suggest the authors try square root scaling or logarithmic scaling in image and video rendering to better handle the big dynamic range and bring more details up.

We do not think that the images/videos are unusually saturated. Rather, what the reviewer perhaps notes might be related to a relatively (but not unusually) low bit depth, which is related to the properties of each sample, and how the PMTs are read out via our DAQ. From here, rescaling the grey values in a non-linear manner, as suggested, would only artificially distort the brightness distribution and thus arguably provide a poorer reflection of the original data.

Minor suggestions

9. In line 521, “Fig. 2f” should be “Fig. 4f”?

Fixed

10. In Fig. 6d, the two images corresponding to 200- μm and 250- μm peak displacement, the central region of the curved plane, illustrated by a red curved line, is already outside the cartoon fish head, but the corresponding 2P images are not void in the center. Please double check.

Fixed

11. In Figs. 8j-k, it is unclear which axes correspond to x-, y-, or z-directions. And why the FOV is labeled to be 3.5 mm in Fig. 8k, but the image is 200 um in dimensions. Please further label and explain in the caption and main text.

The full FOV is 3.5 mm in the example (i.e. nTC₂ configuration), but here we zoomed in to the central 600 microns for the scan. the x dimension in the image corresponds to the x mirror, while the y-dimension is $\sqrt{z^2+y^2}$. This is now explained in the legend.

12. Line 850, PSF tailoring is not truly demonstrated but only proposed in this manuscript, so it should not be claimed in the Conclusion session. Instead, the curved scan profile as reported in Fig. 6 could be mentioned here.

We respectfully disagree. PSF tailoring is demonstrated in two ways. One, as the inevitable consequence of changing the FOV (e.g. Figure 2), and two, also for a given FOV the PSF can be adjusted in size within certain limits by keeping the divergence angle of the laser prior to the back aperture constant while changing the IFP and thus effectively changing beam width. This is explicitly shown in Fig S1j-o. Accordingly, we think it is fair to term it PSF tailoring at this point.

References

1. Xu, C. and W.W. Webb, Multiphoton Excitation of Molecular Fluorophores and Nonlinear Laser Microscopy, in Topics in Fluorescence Spectroscopy: Volume 5: Nonlinear and Two-Photon-Induced Fluorescence, J.R. Lakowicz, Editor. 2002, Springer US: Boston, MA. p. 471-540.

REVIEWER COMMENTS

Reviewer #4 (Remarks to the Author):

For the re-review round, I will simply type my new comments (green-colored) in the context of the old comments (slightly greyed) and the authors' response (*left in black italic, but some key part red-highlighted*). I would like to clarify that I was just trying to help improve the quality of the manuscript, and to defend the reputation of the authors and of Nat Comm. I apologize if some comments sound harsh.

This manuscript reports an interesting exploration towards mesoscale 2PF imaging without the need for a customized and likely oversized objective lens. Instead, an off-the-shelf infinity-corrected objective is used in combination with a non-collimated, diverging input beam so as to extend the focal plane (which could be curved) further from the lens and enlarge the sample-side field of view. Substantial experimental data on zebrafish, mice and fruit flies are provided to demonstrate the applicability of this approach.

While this so-called non-telecentric configuration is an interesting and potentially valuable imaging approach, I have some major concerns, particularly with the claims made by the authors and their quantification of the imaging system's performance. I feel strongly that the authors need to be more objective about their approach and address the following comments with further substantial revisions and/or modifications in order for this manuscript to be acceptable for publication.

Major concerns --- Technology

While this non-conventional usage of an infinity-corrected objective has some merits, the concurrent shortcomings should also be acknowledged and carefully characterized to present an objective and non-overselling picture. Some conclusions claimed in the current manuscript also lack solid theoretical or experimental support, as detailed below:

1. FOV size: throughout the manuscript the FOV of the Zeiss W Plan-Apochromat 20X 1.0 NA objective is quoted to be ~0.5 mm in diameter in a diffraction-limited (DL) sense, which serves as a base FOV against which new configurations are compared. However, since the expanded FOVs in the nTC setups are clearly non-diffraction limited, it seems only fair to compare them to the "usable" ~1-mm-diameter FOV of the 20X objective. Also it is not clear how the DL range is determined to be 0.5 mm here; in Supplementary Fig. S1c, the brightness curve for 0.5 mm DL case seems to be truncated at 0.25 mm radius. I suggest the authors at least extend this curve to the radius where the brightness starts to plunge as the other three cases, and come up with a common criterion, say 50% normalized brightness, for FOV diameter.

We agree that it is possible to get a larger than 0.5 mm FOV with this objective (in fact that is what we show); however in this case, as noted by the reviewer, the system cannot be diffraction limited.

The nTC ideas shown in this manuscript is simply based on sending non-collimated input beam to expand the FOV and the resultant PSFs are mostly not diffraction-limited; so it is

only fair to compare with the limit FOV one can normally obtain using a collimated input using the same objective, no matter whether the PSF is diffraction-limited or not. This is the basic logic of how you set control group in research; I am afraid there is no argument for this. The authors must correct this in order for this to be published.

The 0.5 mm squared FOV is the maximal FOV of the chosen objective on a Sutter MOM DL configuration with 1/e² overfilling of the back aperture (Helmchen and Denk 2005).

*Whether you used it on a Sutter MOM or a customized 2P microscope should NOT affect the diffraction-limited range. But this is not the point here; to be more specific, how large the diffraction-limited FOV is does not matter here. What fundamentally matters, as I said above, is one must perform a fair comparison, **usable FOV against usable FOV**, when publishing at a prestigious journal as Nat. Comm.*

BTW, in an axially symmetric optical system, the maximal FOV should be circular, xx mm in diameter, as the authors stated many times; I don't understand where the "squared FOV" come from here.

The plot noted by the reviewer (S1c) is "truncated" at 0.25 mm radius, because maximum safe mirror displacement has been reached. This is the factory defined "intended" optical configuration of the Sutter-MOM.

Again, however Sutter-MOM restricts the scan range doesn't justify current comparison against the so-called DL range in the manuscript. If that is the limited, and if I were the author, I would simply adapt it or construct a separate 2P microscope to explore how large the usable FOV is when sending in collimated beam. Why? Because not all people use the Sutter-MOM framework; they might buy from Thorlabs, or from Olympus, and in fact, a lot of neuroscientists built their own 2P microscopes.

*Nevertheless, the diagonal of the resultant square FOV is very close to being clipped by the objective back aperture (by factory design), meaning that the strictly diffraction limited FOV of this optical setup **really is ~0.5 mm (or 0.6 mm at most)**.*

Further, much as I hate to point this out, I am afraid I have no choice this time. The diagonal length of a 0.5 mm x 0.5 mm FOV is 0.717 mm, and based on the Fig. (S1d) provided by the author (which I copied below; left panel), there is rarely any hint of vignetting or PSF blurring at the corner of this image, so any educated reader (even without practical experience operating a 2P microscope) will stipulate that the usable FOV is at least 0.717 mm. I don't know how the authors concluded the "strictly diffraction-limited FOV is really 0.5 mm (or 0.6 mm at most)", but the validity of this claim is suspicious at best.

Demonstrating this point more directly would require overriding the maximal safe displacement condition in our mirror commands, which we prefer to avoid because of the risk of damaging them.

.

And again for the second time, the key here is not to argue about how large the DL FOV is; the key is to provide the readers with a fair comparison. The readers deserve to know if an ordinary setup with this 20X objective will be sufficient for their application; they deserve to know better in order to evaluate whether they would proceed to use the nTC idea herein presented. Showing a truncated curve as in Fig. (S1c) (which I copied below, right panel) will only leave the readers to wonder, or even to suspect.

Comment Figure 1. Left: if this is 500 μm in size, then the diagonal length is 717 μm, contradicting the authors' claim of 0.6 mm at most and violating the basic principle of fair comparison. Right: as stated above, please show the full characteristics of the case of collimated input, for a fair comparison.

- PSF measurement with beads: the statement "PSF-dimensions are affected by a myriad of additional factors such as laser wavelength and power, as well as the specific measurement method (e.g. bead types)" (main text, line 318-320) does not comply with basic physics and optics. While PSF measurement, as any other experiments, needs to be carried out with sufficient rigor, the PSF size itself should not depend upon incident power or bead sizes, etc.

Apologies; this was a simple phrasing error. We intended to state that the estimation of PSF dimensions is dependent on all these factors (rather than the actual PSF). We have now clarified that in the MS.

For example, regarding wavelength, the PSF-size (for FWHM) in DL configuration is given in xy and z by (from Zipfel et al. 2003 Nat Biotech):

$$c \quad \omega_{xy} = \begin{cases} \frac{0.320 \lambda}{\sqrt{2} NA} & NA \leq 0.7 \\ \frac{0.325 \lambda}{\sqrt{2} NA^{0.91}} & NA > 0.7 \end{cases} \quad \omega_z = \frac{0.532 \lambda}{\sqrt{2}} \left[\frac{1}{n - \sqrt{n^2 - NA^2}} \right]$$

Where λ is wavelength of the excitation laser, and NA is numerical aperture of

the objective, and n is the refractive index of the sample.

I see what you mean. PSF is indeed a function of wavelength, as well-known, but that is still slightly different from “estimates of PSF is affected by wavelength”. But I will let this go anyway.

Regarding intensity and bead types, we agree that the direct power dependence is small (<1%), however in practical terms the picture is more complex. A “true” PSF size can only be estimated if fluorescence intensity saturation is avoided, and this means that power must be kept minimal – however **this minimal power level is routinely exceeded** during imaging of neuronal structures.

Yes, keeping a low enough power to avoid saturation is standard experimental protocol, and it is doable with waveplate + polarizer + ND filter if needed. But your statement that estimates of PSF are affected by power and bead types implies even if one uses a correct safe power and knowledgeably selected beads, the PSF estimation is still problematic or questionable, which is not the case.

And “minimal power level is routinely exceeded” is not justifiable either. People try their best to minimize the excitation power, no matter single- or two-photon, during imaging experiments to minimize photobleaching and photodamage.

Of course, the degree of this problem strongly depends on the 2P cross-section of the sample (higher cross-section → increased inflation of effective PSF size as measured with FWHM). **Beads have an unknown cross-section**, but it is quite likely to be higher than e.g. GCaMP, accordingly, **bead measurements have a tendency to overestimate the true PSF** compared to typically used biosensors unless power is minimal to avoid the saturation effect.

Again, how bright the beads are is not the key or reason here; one can also lower the power enough. Using small enough beads with low enough power tells enough about the system’s performance.

By “tendency to overestimate the true PSF”, if the authors are talking about the physical size of the beads themselves and theoretically the deconvolution needed, it would be more convincing. But saturation effect is not a real concern in the microscopy community; people all understand they need to control the power low enough to avoid saturation when using beads, and the PSFs estimated in this way are cross-system comparable. PSF is not everything, but it can be measured correctly, and it can be compared.

On the other hand, measuring a PSF using the biosensor itself (which would be the fairest approach) is **not possible** as we cannot generate a “GCaMP-point

source" with current technology. Alternatively, one could measure a relevant biological sample with different power levels and look at z-sectioning. This would give a possibly useful impression of the degree of power-dependence in that specific application, but it would be very difficult to quantify, for example due to the inevitable superposition of excitation planes and also inhomogeneities in the sample.

I am not sure what the authors try to convey here. Just two comments: 1) it is exactly because measuring PSFs with biosensor is hard that people use off-the-shelf beads to estimate PSF. The authors have the right to not like it, but it is physically credible and widely-accepted; 2) I would kindly remind the authors to think twice before using absolute phrase like "not possible" anywhere, in the response or in the manuscript. How about localization-based super-resolution microscopy? How about single-molecule imaging?

Returning to the key points of our study – the demonstration that the FWHM of PSFs as measured with beads grows is a technical side-note, *rather than being specifically linked to the optical performance of our system*. It is only intended as a demonstration of *this often-overlooked problem*. It means that directly comparing PSFs across studies (which we have been asked to do at some depth) is difficult unless all the above parameters (wavelength, intensity, bead type, embedding medium etc.) are known.

*Again, the power-dependence of PSF estimation can be minimized to negligible, if not totally avoided. Yes, comparison of PSFs across studies is not always useful, since PSF doesn't tell everything about the overall imaging performance and the reported PSF values could contain errors. **But this is not my point here, and I didn't ask the authors to compare their PSFs with other studies. All I ask is the authors provide readers with carefully correctly measured PSFs of their systems/setups, and stop thinking other experimentalists do not know how to measure PSFs correctly.***

With the exception of the demonstration of the power dependence, where we went up to >100 mW (as routinely reached when imaging mouse cortex – although of course the CS of biosensors is lower than of beads), we measured all beads *at 15-20 mW which is as close to minimal* as can be achieved while still getting a usable signal. Accordingly, we think that the PSFs are as representative of the system's "true" performance as can be established.

As stated above, people know they should not crank up the power to 100 mW when measuring PSFs with fluorescent beads; there is no need to show Supplementary Figure S1(g-i). I am just trying to help; frankly, keeping talking about this in the manuscript does not reflect well on the authors themselves.

And, I am not sure why the authors need 15-20 mW, for point scanning, to elicit enough signal from fluorescent beads. We used much lower excitation power for beads, frankly speaking (the authors can check other publications too). If that is the reason the authors think saturation is a big concern, I would strongly suggest: 1) increasing the pixel dwelling time to get more signal with even lower power; 2) check the overall beam path to see whether there is an unnoticed reason of collection loss.

- a. Especially, in Supplementary Fig. S1g-i, the PSF sizes were reported to depend on incident power (grows with power, to be more specific). While cranking up the incident power increases the excitable region in the beads phantom (i.e. position further away from the focus gets also sufficient power for 2P excitation), this expanded signal-generating zone is not a PSF. Instead, the PSF dimension is determined only by optics and should be reported with appropriate incident power in a normalized manner (e.g. FWHM). Please do re-check the beads data and re-do PSF measurements if needed.

Please see our response to the above point.

Please see my response above then.

- b. Supplementary notes, line 5-8, paragraph 1, page 1: “the 2P cross-section, is for most fluorescent beads different from that of most popular biosensors for reporting neural activity”. While this statement is true itself, it has little to do with why beads measurements do not tell fully the practical resolution in tissue. It is tissue scattering, not the difference in 2P cross section, that plays a fundamental role in the latter case. Please revise the argument accordingly.

*We agree that scattering is an important aspect. However, the 2P cross section also matters, and it is much more directly related to (mis-)estimates from (usually agarose embedded) beads, which is the subject of this paragraph. As detailed above, in our statement we intended to state that **any differences in the 2P cross section between the beads and e.g. GCaMP can potentially lead to a misleading estimate** of the excitation PSF.*

I don't think I need to be pedantic about 2P signal scales with power squared, with cross section, and with fluorophore concentration, etc, so higher cross section can be balanced by lower excitation power (or stretched pulse). I admire the authors' bravery of challenging existing protocols, but one needs to provide sound proof for such tasks. The reasoning from the authors, I am afraid, does not suffice.

To further clarify this, we have now added the following note at the end of the paragraph in question:

“Notwithstanding, the PSF also depends on the specific scattering in the

sample (see “optical aberrations” below).”

We think that this will usefully guide the reader to the optical aberrations sections, where scattering is discussed at some depth.

OK.

- c. In summary, for an incoherent imaging system, reporting the PSF using small enough beads is theoretically sound and gold standard in practice. It tells the best achievable resolution in the absence of scattering and can be compared across publications (against the authors’ argument in response to review #2, on page 13).

We agree that the PSF measurement using beads is the best available gold standard and we did not mean to imply otherwise. We are in fact using it as such throughout the manuscript.

*We are however simultaneously pointing out that **we think that the excitation wavelength and power, as well as the 2P cross section of the sample also matter**, alongside scattering, and that these can make cross-study comparisons of PSF sizes difficult.*

Again, these factors matter, but they can be correctly set to avoid significant errors, and every well-trained experimentalist knows how to correctly set them. There is no new physics here that people do not understand or haven’t noticed yet. Again, talking about this doesn’t reflect well, sincerely.

*For this to be truly possible, it is important to use **the same beads** and medium, as well as **matched effective laser power and wavelength**, as discussed in the preceding comments. This remains the case even if power is kept low to ameliorate fluorescence saturation.*

As well-accepted in the optics/microscopy field, it is good enough to use small enough beads (i.e. much smaller than theoretical predicted PSF) to estimate PSFs; one can perform deconvolution if the beads are not small enough (although people accept non-convoluted results also).

There is no need to use matched laser power, since collection efficiency and pixel dwell time can vary; using as low power as possible is good enough.

Please, please, do not make arbitrary claims. If the authors don’t believe me, please just consult other trustworthy physicists or optics experts. I don’t know how I could better convince.

- d. The PSF values reported are not quite consistent. In Fig. 1n, the lateral PSF is reported to be ~1.0 μm for nTC1, and ~2.0 μm for nTC2. In Fig. 2b, one can easily estimate that the lateral PSF for the 3.5-mm-FOV case is much larger than 2.0 μm , when comparing to the 500 μm scale bar. Copying this image into ImageJ, and the lateral FWHM measures ~15 μm at least. This discrepancy should be addressed.

The scan profiles estimated by the camera approach are not intended to provide numerical information on the real PSF sizes, as this will almost certainly be a gross overestimate. Rather, the goal of the camera approach was to establish a simple way to visualise how PSFs vary over the full field of view (e.g. field curvature, tilting etc, and how scans distribute across space in the varying configurations shown throughout the MS). This is now explicitly noted in the text, and we thank the reviewer for drawing our attention to this source of possible confusion:

“However, the resultant scan profiles overestimated PSF sizes and were thus not suitable to determine their absolute dimensions – rather, the goal was to observe relative variations in PSF shapes, positions and orientations over the full FOV.”

Thanks for addressing this.

- e. Related to this point is that reviewer #3 asked how different the PSF is off axis at peripheral locations, but the authors answered only about axial PSF. It is equally important to characterize the degradation in lateral PSF at the periphery of the FOV, especially considering that this unconventional use of a commercial objective breaks down the Abbe sine condition according to which commercial high-NA objectives are designed. As already requested by Reviewer #1, I would also urge the authors provide careful ZEMAX simulation and/or bead-based PSF characterization throughout the FOV for each nTC setup.

We now provide new experimental data to directly quantify PSF variations over the full field of view for all configurations (new Figure 2c-e). This directly shows that while the PSFs do indeed tilt a little at the edge (as already shown by the camera approach), they are not strongly distorted relative to the centre in any configuration. We hope that the new data noted above fully addressed this concern.

Thanks for addressing this.

3. For PSF measurement from the side using a 4X 0.2 NA air-objective and a camera (Fig. 2a-b): the caveat is that the smallest that can be resolved by this approach is inherently limited by i) the theoretical lateral resolution of the 0.2 NA objective; ii) aberration induced by the complicated water-plastic cuvette wall-air interface (no info on how thick the water and plastic layers are); iii) sampling density determined by tube lens and camera pixel size (no details given here); and iv) the limited depth-sectioning capability (or axial resolution) of this essentially wide-field measurement. Therefore, the accuracy of this approach is definitely insufficient for the DL case, and the extent to which it affects those nTC cases mandates further consideration and careful characterization.

We agree that the camera generated scan profiles are not suitable to provide a quantitative measurement of absolute PSF dimensions. This is done by the

beads (including now also the new added data beneath). We now discuss this further in the associated text:

“However, the resultant scan profiles overestimated PSF sizes and were thus not suitable to determine their absolute dimensions – rather, the goal was to observe relative variations in PSF shapes, positions, and orientations over the full FOV”

And again in the Methods section on PSFs:

“Note that this technique as implemented here tended to overestimate PSF sizes (e.g. due to the low N.A. objective, and because the cuvette was slightly vertically tilted relative to the camera axis to prevent back-reflections) and was therefore not suitable to determine absolute PSF dimensions – rather it was intended to enable relative comparisons of PSFs across scan configurations and positions. All quantitative PSF measurements were instead performed on beads (see above).”

Thanks for addressing this.

4. Signal enhancement: at least two types of signal boost were discussed in the manuscript, i.e. “nTC can yield an overall signal boost of up to ~18-fold with an above-stage detector (or ~4-5-fold at equal laser power on the sample)” (line 217-219, main text). The logic here is the under-filling associated with nTC setups brings a higher percentage of laser power to the sample (as reported in Fig. 1j) and thus “this signal loss was greatly outweighed by the increased overall availability of photons for collection in the first place (relative to DL: ~10-20-fold for nTC1 and 16-18 for nTC2 and 2)” (line 424-426, main text). There are several questions that I suggest the authors to explain:
 - a. The improved power percentage to the sample, albeit important in practice for deep 2P imaging, should not be factored into the calculation of signal boost. It is more meaningful and fair to compare signal for given incident power onto the sample, since eventually it is photo- and thermal damage in the sample, not in the objective lens, that limits the highest power to use.

We agree that total power does not necessarily directly translate into a better signal. Accordingly, we also agree that this ~4-fold power gain is not necessarily part of the signal boost per se. This is why we explicitly note that signal boost is 4-5-fold if laser power is kept constant (rather than up to 18-fold) in our initial bullet point list. Moreover, when discussing reduced power loss, we note that this is potentially helpful if the goal is to image deep, or if multiple setups share a single laser.

Of course, as the PSF grows e.g. by using different nTC configurations, the power density experienced by a given image structure also changes, which makes it possible to increase laser power further before the sample “starts to burn”. Accordingly, the relationship between the best signal/noise that can be achieved and laser power / PSF shape is complex. We further elaborate on this point below.

Good to see we agree on this point. Thanks.

- b. More importantly, the concept that an enlarged excitation volume generates more signal is questionable. Based on fundamental principles of 2P excitation, when the effective focusing NA decreases by M-fold and thus the lateral focal area growing by M^2 -fold, the excitation beam intensity decreases by $1/M^2$, and thus the 2PE efficiency decreases by $1/M^4$. When focusing into an infinite thick uniform sample (e.g. fluorescein solution cuvette), this $1/M^4$ deficiency in excitation efficiency can be compensated by the correspondingly expanded focal volume, i.e. M^2 expansion in lateral area and M^2 elongation of axial range (see Chris Xu [1]). But for real-world biological tissues, especially sparsely labeled neurons, the biosensor distribution is clearly not uniform, and therefore the significantly expanded axial PSF (up to 40 μm) in the nTC setup might NOT encounter biosensor and generate 2P signal across the entire axial range, and therefore yield less 2P fluorescence. Therefore, how the nTC setups, despite its collection loss, yield overall signal boost “even if keeping effective laser power on the sample constant” (line 431-432) is unclear and unconvincing. I urge the authors conduct rigorous experiments (e.g., imaging into the same fluorescein solution with identical pixel dwelling time, and compare the obtainable signal per pixel area) to quantitatively prove the extent of signal boost. Otherwise this claim should be removed.

We agree that how a given PSF permits us to capture a given image structure (e.g. a cell body) depends on both the structure and the PSF, such that e.g. part of a large PSF will miss the structure and therefore the signal will stop increasing. Conversely, however, a large PSF is also much more likely to partly or fully capture a structure in the sample at all. Clearly this is a non-trivial relationship, and the real signal boost (or loss) will depend on the application. This is part of the reason why we provide so many different example tissues and applications for the different configurations, alongside the direct quantification based on fluorescein-soaked tissue shown in the supplement (which seems very similar to the experiment suggested by the reviewer) as well as the direct comparison of the same set of neurons in the case of the zebrafish spinal cord. The latter, for example, shows that the likely most experimentally useful configuration for this particular sample is around nTC1 (1.8 mm) rather than the largest possible nTC2 configuration.

Just to clarify, the fluorescein-soaked tissue is nowhere similar to what I suggested. The data shown in Fig. S3d, as stated below, is what I suggested. Plus, fluorescein-soaked tissue, even carefully sealed without vaporization over time, is not a quite good phantom, as the scattering and absorbing properties could be quite different from biological samples. People have established several types of scattering phantoms, made from e.g. non-fluorescent beads, or TiO_2 , or intralipid, etc. Again, for those well-known approaches, it is better to follow standard protocols.

Nevertheless, we addressed the reviewer's concern and now also performed the suggested experiment. Specifically, we imaged the central point of each FOV in a cuvette filled with low concentration fluorescein solution (to not saturate the PMTs). **The laser was kept constant at the level of the scan mirrors** (hence more laser on the sample, as discussed), and the PMT gain and dwell time were also kept constant. We then compared the brightness readings from each of the two detectors for each nTC configuration against the readings from the DL configuration. This further confirmed that all nTC configurations provided a substantial overall signal boost on both detectors. This is now added to Figure S3d.

Thank the authors for making new data regarding my comments.

Unfortunately, as stated in the red-highlighted part above, the power was maintained constant at the scan mirror. According to Figure 1j, the scan mirror-to-sample transmission is only 10% in the so-called DL case, but ~35-40% for all nTC cases. This means the power on sample for the DL case is around one fourth of the nTC cases, thus the 2P excitation efficiency will be only 1/16.

Therefore, once more, this is not a fair comparison; it is the power on the sample that truly matters. If we divide the signal ratios in Fig. S3d by 16, would the author agree that the signal per excitation power in the nTC cases are actually significantly lower? Would the authors agree with my original comment that "the concept that an enlarged excitation volume generates more signal is questionable"?

A related question one can help but ask is, were the data shown in Fig. S3(b-c) also acquired with constant power on the galvo mirror, thus severely different power on the sample? If yes, would the authors please re-phrase their claim about the enlarged excitation volume generating more signal?

Plus, for in vivo experiments, usually it is only possible to collect the epi-fluorescence (i.e. "Above-stage" case in Fig. S3d).

In summary, I respectfully request the authors re-reconsider the data and re-think carefully whether the signal growth seen here is of practical usefulness, and how to better present these observations to avoid misleading biologists who are not so familiar with two-photon physics and instrumentation.

- c. One caveat is that the authors are actually describing the final image brightness, where the scan step (i.e. digital pixel) is much finer than the lateral PSF so that the collected photons (i.e. the reported brightness) at each digital pixel is essentially photons from all neighboring digital pixels binned together. I would like to emphasize that: i) such enhancement in pixel brightness is not equal to signal boost, as the latter typically referred to 2P fluorescence intensity, i.e. photons per cross-sectional area per sec; ii) it is the emission intensity, not the emission power per se, that matters in practice; otherwise, one can also digitally

bin a diffraction-limited image to boost the brightness in the cost of a blurred image. Please double-check and make the concept clear in the manuscript.

Please see previous response.

The authors didn't really understand my comment, I am afraid; and the response to my previous comment didn't touch my question here. To be more specific, the data in Fig. S3(c-d) are all photon counts (i.e. energy, or power given the dwell time is constant), but what I am asking/emphasizing is emission intensity, i.e. emission power per given pixel area. Please re-think about my comment and re-response.

5. Aberration: as pointed out by Reviewer #1 already, no “new types” of aberration does not mean “no more aberrations”. **There are always 5 major types of monochromatic aberrations**, and it is the amount of aberration that matters. So I suggest the authors re-phrase their argument about aberration type on line 817-818 in the main text. And,

*We agree, however the comment on “no new aberrations” stems from a previous round of reviews, and we feel it is important to explicitly note that **new types of aberrations do not arise**. Moreover, beyond the PSF expansion, which we think is extensively documented and discussed, we do not see any notable worsening in other types of aberrations (e.g. also see the new data on PSFs across the full FOV noted above, point 2e). Accordingly, we have rephrased as follows:*

*“In general, beyond the PSF expansion that results from bypassing the objective’s infinity correction (Figs. 1,2, Fig. S1), the change from a standard 2P DL-setup to an nTC configuration **does not bring about new types of aberrations, nor does it notably worsen the inevitably pre-existing aberrations from the DL configurations**”.*

As stated in my previous comment (bolded here), there is no new physics about aberration theory, and no matter how one alters his or her point-scanning microscopes, the types of aberration are always spherical, coma, astigmatism, field curvature, distortion, and chromatic aberrations. There is no need to say “does not bring about new types of aberrations”; on the contrary, again, claiming this reflects poorly on the authors. Please, please, consult some trustworthy physicists if the authors don't believe me.

And the definition of diffraction-limited PSF already implies no aberration, or in practice, negligible aberrations. The PSF expansion itself already implies increased amount of aberration; one prominent example is the significantly increased field curvature shown in Fig. 1c and Fig. 2b. So the claim that “nor does it notably worsen the inevitably pre-existing aberrations ...” has no leg to stand on.

Please, please, don't make arbitrary claims about well-established fundamental optics and physics. Again, again, please consult trustworthy physicists.

- a. In line 852, “reduced spherical aberrations” was claimed in the Conclusion part which is, however, against the common sense that spherical aberration grows with defocusing. It is possible that in such nTC setups, the underfilling of back aperture reduces the effective entrance pupil size and subsequently the overall spherical aberration, but detailed quantitative evidence is needed to support this claim.

We followed the reviewer's suggestion and have removed this statement on spherical aberrations.

Thank you.

- b. Besides field curvature, another expectable important aberration is distortion in FOV, as also mentioned by Reviewer #2. Note that we are not talking about distorted PSF, but distortion aberration regarding how much the lateral magnification changes for a given focal plane. Please reconsider the question and provide clear, rigorous characterization.

We have now measured this on a fixed pollen sample by taking identical scans with different stage xy translations, followed by direct superposition of a corresponding image region. No evidence of distortion was observed. The data is added as Figure S2:

Thanks for performing the characterization. The data look convincing.

6. Finally, the term “non-telecentric” is not a very appropriate here either. The reason is that telecentricity refers to collimated chief ray and requires relaying the galvo (using scan and tube lens) to exactly the back focal plane of an objective. In standard 2P microscopes, however, the galvo is often relayed to the back aperture, rather than the back focal plane, for maximal power throughput. Therefore, common 2P microscopes are already non-telecentric. So I suggest the authors rethink about this and probably choose a better term.

Thank you for giving us the opportunity to clarify this point. We agree that some 2P setups are designed to use a mildly non-telecentric beam for the aforementioned reasons. However, this usually comes about with a rather small angle of divergence where the goal, as noted, is to increase effective power available for imaging. Importantly, this is, by definition, no longer a diffraction limited system, as this requires the beam to be parallel (for an infinity corrected objective).

Here, we think that we have gone substantially beyond such systems, for example by explicitly varying the degree of divergence to much larger angles

and quantified the consequences in depth. As such, we think that the term non-Telecentric usefully summarises our efforts, and we have thus opted to keep it.

Thank the authors for the response. I agree with them.

Major concerns --- Applications

7. The top concern is that the PSF can grow up to 40 μm in the axial direction, which is not truly “single-cell resolution”, as evidenced by the final images in Figs. 4-9 and pointed out by previous reviewers. The ability to see some individual isolated cells in sparsely labeled specimen is not equivalent to single-cell resolution. Again, the approach reported in this manuscript has its unique merits for cost-effective, mesoscale 2P imaging, but I suggest the author avoid overselling it as a “single-cell resolution” imaging solution.

Following the reviewer’s advice, we have now removed the reference to single cell resolution from the title. In the abstract and throughout the text we have either removed this note, or where we have kept it, made clear that this depends on the specific sample.

Thank the authors for addressing accordingly.

8. Another big concern is a lot of images and videos shown here are too saturated (especially, Fig. 5d, 5k, 7d and associated supplementary videos). I suggest the authors try square root scaling or logarithmic scaling in image and video rendering to better handle the big dynamic range and bring more details up.

*We do not think that the images/videos are unusually saturated. Rather, what the reviewer perhaps notes might be related to **a relatively (but not unusually) low bit depth**, which is related to the properties of each sample, and how the PMTs are read out via our DAQ. From here, rescaling the grey values in a nonlinear manner, as suggested, would **only artificially distort the brightness distribution** and thus arguably provide **a poorer reflection** of the original data.*

The DAQ cards mentioned in the manuscript are PCIe-6363 and PCI-6110 from National Instruments; the former has 32 AI (16-Bit, 2 MS/s), and the latter has 4 AI (12-Bit, 5 MS/s/ch) according official specifications. Both are pretty good bit depths. I don’t know what the authors mean by low bit depth.

The figures I mentioned look quite saturated to me (and to some of my colleagues, so probably to some other readers in the future), and I am merely suggesting common approaches in literature to better showing high dynamic range data. The “artificial distortion” sometimes helps, and does not “provide a poorer reflection”.

The authors have the right to choose. Again, I am just trying to help.

Minor suggestions

9. In line 521, "Fig. 2f" should be "Fig. 4f"?

Fixed

Thanks

10. In Fig. 6d, the two images corresponding to 200-um and 250-um peak displacement, the central region of the curved plane, illustrated by a red curved line, is already outside the cartoon fish head, but the corresponding 2P images are not void in the center. Please double check.

Fixed

Thanks

11. In Figs. 8j-k, it is unclear which axes correspond to x-, y-, or z-directions. And why the FOV is labeled to be 3.5 mm in Fig. 8k, but the image is 200 um in dimensions. Please further label and explain in the caption and main text.

The full FOV is 3.5 mm in the example (i.e. nTC2 configuration), but here we zoomed in to the central 600 microns for the scan. the x dimension in the image corresponds to the x mirror, while the y-dimension is $\sqrt{z^2+y^2}$. This is now explained in the legend.

Thanks.

12. Line 850, PSF tailoring is not truly demonstrated but only proposed in this manuscript, so it should not be claimed in the Conclusion session. Instead, the curved scan profile as reported in Fig. 6 could be mentioned here.

We respectfully disagree. PSF tailoring is demonstrated in two ways. One, as the inevitable consequence of changing the FOV (e.g. Figure 2), and two, also for a given FOV the PSF can be adjusted in size within certain limits by keeping the divergence angle of the laser prior to the back aperture constant while changing the IFP and thus effectively changing beam width. This is explicitly shown in Fig S1j-o. Accordingly, we think it is fair to term it PSF tailoring at this point.

With all due respect, this is not why PSF tailoring typically means in the optics/microscopy field. Please consult respectable optics experts to decide whether to keep this term/claim.

For your reference, I provide two examples of what the community typically accept as PSF tailoring here:

*Song A, Charles AS, Koay SA, et al. Volumetric two-photon imaging of neurons using stereoscopy (vTwINS). Nat Methods. 2017;14(4):420-426.
doi:10.1038/nmeth.4226*

Yi Xue, Kalen P. Berry, Josiah R. Boivin, Christopher J. Rowlands, Yu Takiguchi, Elly Nedivi, and Peter T. C. So, "Scanless volumetric imaging by selective access multifocal multiphoton microscopy," Optica 6, 76-83 (2019)

References

1. Xu, C. and W.W. Webb, *Multiphoton Excitation of Molecular Fluorophores and Nonlinear Laser Microscopy*, in *Topics in Fluorescence Spectroscopy: Volume 5: Nonlinear and Two-Photon-Induced Fluorescence*, J.R. Lakowicz, Editor. 2002, Springer US: Boston, MA. p. 471-540.

There appear to be five outstanding areas of discussion from the previous round of reviews.

1. The size of the diffraction limited FOV (related to previous Figure S1C)
2. How to measure/discuss PSFs (related to previous Figure S1G-I)
3. Signal boost under nTC (related to previous Figure S2D)
4. A discussion on aberrations
5. The note on PSF tailoring in the conclusion paragraph

To streamline the process going forward we here do not provide a point by point but rather deal with each point as one.

1. The reviewer notes that the FOV of the x20 Objective used is larger than 0.5 mm in diameter. Here, as explained previously, the 0.5 mm is the total effective size of the FOV when the objective is mounted under the commercially available Sutter MOM. This is because the galvanic mirrors are positioned such that they would run into each other if they were driven to guide the beam centre beyond the edge of the objective's back aperture which would then reveal the 'true' FOV of the objective. Clearly, overriding these safe mirror commands to get the requested curve is not an option.

Alternatively, it would principally be possible to modify the optical pathway to allow greater absolute beam travel from the same mirrors while keeping the beam collimated, thus maintaining a DL configuration that uses the objective's full FOV. However, this would require a complete overhaul of the current beam-path.

As another alternative still, we therefore axially offset the scan mirrors to no longer be centred relative to the objective centre, but rather to catch one edge:

This in turn allowed us to measure the requested signal drop at the objective edge: (curve labelled DL*), which is now added to SFig1c, as requested:

Based on this estimate, the objective's full FOV under a DL configuration is approximately 0.7 mm. We have now updated the "0.5" to "0.7" throughout the MS, and explained this situation in the main text:

"An off-the-shelf infinity-corrected galvo-galvo Sutter-MOM setup equipped with a 20x objective (Zeiss Objective W "Plan-Apochromat" 20x/1.0) offers a square FOV diameter of ~0.5 mm (left in Fig. 1g). This can be principally extended to a round ~0.7 mm diameter usable diffraction limited (DL) FOV of the objective (see below), for example by redesigning the native beam path to allow a greater range of laser travel"

And, in the legend to Figure S1:

“The two traces for the DL configuration denote the brightness over the ‘native’ 0.5 mm square FOV of the Sutter MOM equipped with the used Zeiss x20 Objective (DL), and of the total usable FOV of the objective ‘in isolation’ (~0.7 mm, DL), here measured by artificially offsetting one of the scan mirrors such that the scanning laser beam could go beyond the edge of the objective back-aperture.”*

2. The reviewer argues that PSFs can be reasonably compared across studies if laser power is kept low. We agree. This is why presented PSFs are measured at “low” 15-20 mW power. In the previous MS, we also showed how the silhouette of the PSF grows with increased power – as discussed, this effect is linked with fluorescence saturation at the PSF centre (previous SFigure 1g-l). In hand, we also noted that PSF measurements are not wavelength invariant, showing measurements taken at both 927 and 960 nm. This data, and associated discussion, was included in response to a previous round of reviews at another journal, where we were asked to numerically compare our PSFs with those of several previously published papers. Several of these papers did not list laser power, wavelength and/or bead types. To make our PSFs forward compatible, we presented this information alongside these plots.

However, we agree that the data is not necessary to support any of our central claims, and we have now removed it. We also removed the associated discussion paragraph in the supplement.

In addition, the reviewer commented that even 15-20 mW, if anything, might be considered “high” power, implying that our presented PSFs might already begin to show fluorescence saturation. We do not think this is the case. This is for example based on new data that directly compares the PSF from a single bead taken at 2 different power levels (nTC₂, 1.2 configuration):

The size of the PSFs measured at the two power levels, as expected, is identical within the measurement accuracy. For simplicity, we do not include this new data in the MS.

For comparison, while many recent 2P method papers do not list laser power used for obtaining PSFs, Sofroniev and colleagues (eLife) do mention the power used for characterising the well-known mesoscope. In this case they state in the associated figure legend: “*The excitation intensity was 8.75 mW, below excitation saturation of the beads*”. Accordingly, in this well-known reference paper, the authors used a laser power that is nearly an order of magnitude above the very low power PSF we provide in the above (1 mW), which is indistinguishable in apparent size from the same bead measure at 23 mW. Accordingly, the 15-20 mW used to characterise PSFs in our paper are in line with available literature.

- The reviewer commented on our approach of quantifying possible signal boost under nTC. Here, two main effects combine. One, since we do not overfill the objective back aperture, a higher proportion of laser can reach the sample, thus giving an increase in achievable brightness. Two, the larger PSFs will increase excitation, so long as there is something to excite within the excitation volume. As a result, the ‘true’ signal boost will depend on the application.

Previously, we quantified the achievable signal boost in two ways – by imaging fluorescein-soaked tissue (mimicking a spatially heterogeneous sample SFig. 3A-C), and again by sampling a more spatially homogenous fluorescein solution (SFig. 3D). Regarding the latter, the reviewer argues that power should be kept constant on the sample, rather than on the mirrors as we had opted to do. In this way, the data would be informative about the signal boost associated with the PSF expansion alone, rather than a combined effect of reduced laser loss and PSF expansion.

While we think that either configuration is reasonable and useful, for simplicity, we now provide the data from the requested configuration. As expected, the overall signal boost is somewhat lower than for the combined effect, now coming in at ~2-3 fold signal boost relative to DL depending on the nTC configuration and detector path (previously ~2-5 fold). Nevertheless, for all nTC configurations, the signal remained boosted. This data now replaces previous Figure S3D:

- The reviewer suggests a rephrase of our discussion on aberrations in lines 817-818. Specifically, they note that pointing out that “no new types of aberrations arise” is superfluous. We agree that this is the case, and again, the brief note was there in response to a previous round of reviews.

In particular, the reviewer comments on PSF expansion and field curvature. We agree. In fact, the PSF expansion was explicitly noted in the first part of that sentence, and we have now added a second note on the field curvature. We also removed the note that no new types of aberrations arise:

“In general, beyond the PSF expansion that results from bypassing the objective’s infinity correction (Figs. 1,2, Fig. S1), and the increased field curvature that inevitably follows from FOV expansion, the change from a standard 2P DL-setup to an nTC configuration does not notably worsen other types of aberrations.”

5. Finally, the reviewer argues that PSF tailoring should not be included in the conclusion paragraph. While we continue to find this assertion debatable, we have now removed this non-essential note.

REVIEWER COMMENTS

Reviewer #4 (Remarks to the Author):

In the last round of review, I wrote detailed comments, color-coded, on all the authors' rebuttal, in a 16-page PDF, with the hope that they would be of help to the authors to make the manuscript better. The response I received on this round, surprisingly, is only 5 pages, and the authors reluctantly chose, on purpose or randomly, 5 comments to address, and ignored all other comments I curated.

PLEASE RE-WRITE your response by addressing all comments I gave last time.

Secondly, for the comments they chose to address this time, they didn't seem to provide fully convincing responses for all the them. In specific:

1. Regarding diffraction-limited FOV, they still made excuses about why they cannot scan over 0.7 mm — "This is because the galvanic mirrors are positioned such that they would run into each other if they were driven to guide the beam centre beyond the edge of the objective's back aperture". Scientifically, why should I or other readers care about how you are limited by Sutter? More importantly, the newly measured curved, shown as the dashed line added to SFig1c, appears to be identical to the blue line I drew randomly for illustration in the last round of review (except a flat extension beyond 0.5 mm radius). DID YOU REALLY MEASURE IT?!

2. OK, it seems the authors finally agreed with me there is no need to emphasize PSF measurement should not use too high power. Thanks. Logically, the example eLife paper saying 8.75 mW is below the saturation limit does NOT really prove the 23 mW power you used is also below the limit, right? But anyway, I now trust you that for your system, 23 mW doesn't seem to saturate the beads.

3. I will first ignore the authors' quibbles about "either configuration is reasonable and useful" — commercial Ti:S lasers have enough power to compensate your loss in nTC cases. It is the power on sample that is of importance always.

I would like to ask the authors to compare this new Fig. S3D with the old one, and answer my comment last time: "According to Figure 1j, the scan mirror-to-sample transmission is only 10% in the so-called DL case, but ~35-40% for all nTC cases. This means the power on sample for the DL case is around one fourth of the nTC cases, thus the 2P excitation efficiency will be only 1/16. ... If we divide the signal ratios in Fig. S3d by 16, would the author agree that the signal per excitation power in the nTC cases are actually significantly lower? Would the authors agree with my original comment that "the concept that an enlarged excitation volume generates more signal is questionable"?"

Just in case the authors are confused, my concern is your transmission data implies that the improvements shown in the old Fig. S3D should be divided by 1/16 to get the new Fig. S3D. Please address the discrepancy between experiments and physics theory. THIS ONE IS CRITICAL, as I said many times, there should no false claims to make in a Nat Comm level paper.

4. I am glad the authors decided not to mention new types of aberrations. Many thanks.

5. OK. I don't want to debate about this anymore either. Thanks.

Reviewer #1 (Remarks to the Author)

I agree that the authors' idea of diverging the beam to enlarge the FOV seems interesting if people want a quickly large-scale screening under the 2P microscope without changing the objective.

However, after reading Review 4's comments, the authors' rebuttal letter, and the current version of the manuscript, I realized that the problems existing in this manuscript are still so significant that I could not recommend it be accepted unless they are solved. Based on points 1 and 3 of reviewer4, I added additional comments:

"Signal boost" issue (related to point 3 from reviewer 4):

The authors claimed that their method could give a "signal boost," which increased the brightness of their recordings $>10\times$ diffraction-limited configuration. First, it does not make sense to claim that signal is increased because more laser power is delivered to the sample. As reviewer 4 mentioned, the laser sources are powerful enough to compensate for the loss in the light path, so this is not something important worthy to mention.

We agree that lasers can principally compensate. However, reducing laser-loss in the beam path is still useful, for example when multiple setups share a single laser.

Again, the authors claimed that even keeping the laser power on the sample constant, they still observed this "boost," which conflicted with the well-established 2p excitation theory. In the last round, both another reviewer and I pointed out that Chris Xu, and Watt Webb (inventors of 2p and 3p microscopes) proved that if the sample is uniformly distributed in the excitation volume, the total photon excited is constant with whatever NA or PSF profile you used. For the complete mathematical derivation, please check page 479 in the documents I attached.

So, if we all agree that the collection efficiency in nTC1 and nTC2 are lower than in DL condition, then the authors will observe a similar or lower signal in nTC1/2 compared to DL configuration, but not higher. Therefore, authors must explain why their claim and results (Fig.S3d) conflict with the theory. One guess is that this "boost" only exists when you image an ununiform sample like the fluorescein-soaked tissue.

We believe that this point comes down to miscommunication about what is meant by boost. To sidestep this issue, we have now removed the entire section.

As a brief explanation – we were referring to the usually brighter images that are obtained when the elongated PSFs catch more structure in a heterogeneous sample – much like a Bessel beam as the reviewer notes in the below. To what extent this is a good or a bad thing of course depends on the sample and experimental question.

By coincidence or other reasons, their DL imaging did not focus on any sample. However, it isn't very sensible since you can always move your focal plan to find the sample.

Our DL configuration examples focus on the sample at the same central focal plane as the other examples shown. As the reviewer notes, the non-uniformity in the sample serves to optically merge more structure in the nTC configurations, which is why they appear brighter.

Another reason may be that the low z resolution of the nTC configuration has the effect of summing more planes together, similar to the Bessel beam, which could increase the chance of finding samples. However, this is nothing about "signal boost", because it does not increase the SNR, but add lots of background instead.

We agree that the elongated PSFs of nTC do essentially work like a small Bessel beam.

Also, this "understage detection"(Fig.S3a) is quite confusing because I do not think anyone could use this method to do practical recording, especially since the targets are neuroscience.

We agree that for imaging a large sample like the in vivo mouse cortex, substage collection does not help.

However, substage detection is common practice in many labs that work on samples that are sufficiently small for some of the light to pass through the sample. Examples work on larval zebrafish and Drosophila (unless there is too much cuticle beneath the sample which depends on the experimental approach), essentially all retina labs (retina itself is inherently transparent, and is routinely flat-mounted with pigment-epithelium removed), and researchers working on cultures or slices (sometimes it is useful to not work with widefield illumination in these cases, for example for optogenetics).

Nevertheless, since we removed the section on image brightness / "boost", all arguments around substage collection are also removed.

All in all, all claims related to "signal boost" in this manuscript are against common sense, and the results are not convincing. I would recommend to the authors to either remove this concept in the paper or give the more solid mathematic proof (which I believe is very difficult for the authors).

We have removed this section.

"Resolution" issue (also related to point 1 from reviewer 4): In my opinion, the authors overclaimed the quality and resolution of their technique, especially in nTC2. If we believe the PSF test in Fig.1j to 1 o is correct. Then nTC2 can NOT achieve single-cell resolution, at least in the dense sample such as Zebrafish and mouse brain. Authors must clarify this in the manuscript. Otherwise, people will be misled, spending lots of time testing something theoretically that can not give what they want.

The claim of single cell-resolution had been removed in a previous version, but it appears we missed a single instance, which is now also removed.

Also, I did not buy that the authors achieved an "adjustable ("PSF tailoring") system's 3D PSF". To my understanding, this is just another way to say that the system has a bad resolution, but they can make it even worse.

We have removed the section on tailoring.

In general, I strongly suggest the authors tune their tone down, remove claims including 1 signal boost, 2 PSF tailoring, and 3 single-cell resolution in nTC2 (3.5mm FOV), and reorganize the manuscript focusing more on the general applicability perspective (give examples with different objectives, and different 2p systems).

We have removed all noted points.

Reviewer #4 (Remarks to the Author):

In the last round of review, I wrote detailed comments, color-coded, on all the authors' rebuttal, in a 16-page PDF, with the hope that they would be of help to the authors to make the manuscript better. The response I received on this round, surprisingly, is only 5 pages, and the authors reluctantly chose, on purpose or randomly, 5 comments to address, and ignored all other comments I curated. PLEASE RE-WRITE your response by addressing all comments I gave last time.

We have taken editorial guidance to not respond to this request.

Secondly, for the comments they chose to address this time, they didn't seem to provide fully convincing responses for all the them.

In specific:

1. Regarding diffraction-limited FOV, they still made excuses about why they cannot scan over 0.7 mm — “This is because the galvanic mirrors are positioned such that they would run into each other if they were driven to guide the beam centre beyond the edge of the objective's back aperture”. Scientifically, why should I or other readers care about how you are limited by Sutter? More importantly, the newly measured curved, shown as the dashed line added to SFig1c, appears to be identical to the blue line I drew randomly for illustration in the last round of review (except a flat extension beyond 0.5 mm radius). DID YOU REALLY MEASURE IT?!

Yes, we did measure it:

The source data for the curve, plus the others shown above, is now available on Zenodo (<https://zenodo.org/record/5599524#.YXfrxhwo9hE>) which is also where we will be uploading all other relevant plots upon publication.

2. OK, it seems the authors finally agreed with me there is no need to emphasize PSF measurement should not use too high power. Thanks. Logically, the example eLife paper saying 8.75 mW is below the saturation limit does NOT really prove the 23 mW power you used is also below the limit, right? But anyway, I now trust you that for your system, 23 mW doesn't seem to saturate the beads.

No further comment.

3. I will first ignore the authors' quibbles about "either configuration is reasonable and useful" — commercial Ti:S lasers have enough power to compensate your loss in nTC cases. It is the power on sample that is of importance always.

I would like to ask the authors to compare this new Fig. S3D with the old one, and answer my comment last time: "According to Figure 1j, the scan mirror-to-sample transmission is only 10% in the so-called DL case, but ~35-40% for all nTC cases. This means the power on sample for the DL case is around one fourth of the nTC cases, thus the 2P excitation efficiency will be only 1/16. ... If we divide the signal ratios in Fig. S3d by 16, would the author agree that the signal per excitation power in the nTC cases are actually significantly lower? Would the authors agree with my original comment that "the concept that an enlarged excitation volume generates more signal is questionable"?"

Just in case the authors are confused, my concern is your transmission data implies that the improvements shown in the old Fig. S3D should be divided by 1/16 to get the new Fig. S3D. Please address the discrepancy between experiments and physics theory. THIS ONE IS CRITICAL, as I said many times, there should no false claims to make in a Nat Comm level paper.

The data and associated discussion around the signal boost have been removed. We will provide no further comment on this point.

4. I am glad the authors decided not to mention new types of aberrations. Many thanks.

No further comment.

5. OK. I don't want to debate about this anymore either. Thanks.

No further comment.

REVIEWERS' COMMENTS

Reviewer #1 (Remarks to the Author):

The authors have addressed all my questions and I have no more requirements for them.

Reviewer #4 (Remarks to the Author):

No further comments